# Extracting the femtometer structure of strange baryons using the vacuum polarization effect

The BESIII Collaboration*

One of the fundamental goals of particle physics is to gain a microscopic understanding of the strong interaction. Electromagnetic form factors quantify the structure of hadrons in terms of charge and magnetization distributions. While the nucleon structure has been investigated extensively, data on hyperons are still scarce. It has recently been demonstrated that electron-positron annihilations into hyperon-antihyperon pairs provide a powerful tool to investigate their inner structure. We present a method useful for hyperon-antihyperon pairs of different types which exploits the cross section enhancement due to the effect of vacuum polarization at the $J/\psi$ resonance. Using the 10 billion $J/\psi$ events collected with the BESIII detector, this allows a precise determination of the hyperon structure function. The result is essentially a precise snapshot of the $\bar{\Lambda}\Sigma^0$ ($\Lambda\bar{\Sigma}^0$) transition process, encoded in the transition form factor ratio and phase. Their values are measured to be $R = 0.860 \pm 0.029(\text{stat.}) \pm 0.015(\text{syst.})$, $\Delta\Phi_{\bar{\Lambda}\Sigma^0} = (1.011 \pm 0.094(\text{stat.}) \pm 0.010(\text{syst.}))\,rad$ and $\Delta\Phi_{\Lambda\bar{\Sigma}^0} = (2.128 \pm 0.094(\text{stat.}) \pm 0.010(\text{syst.}))\,rad$. Furthermore, charge-parity (CP) breaking is investigated in this reaction and found to be consistent with CP symmetry.

One distinctive feature of the strong nuclear interaction and a prerequisite for our existence is the confinement of nearly massless quarks into stable and massive hadrons such as protons or neutrons that constitute the matter we are made of. A coherent understanding of the dynamics of the strong interaction, however, remains one of the most intriguing puzzles of physics. The main challenge is the very nature of confinement: the quarks and gluons cannot be observed as bare particles, but are dressed by the strong interaction into quasi-particles, or constituent quarks, that form the bound systems we know as hadrons. The distribution and motion of quarks inside hadrons is quantified in terms of, e.g., electric and magnetic form factors ($G_E$ and $G_M$), which offer an empirical tool to study the strong dynamics. The proton, as the most stable composite particle we know, with a lifetime much longer than the age of the Universe, offers an excellent testing ground for the strong interaction. The space-like form factors of the proton have been the subject of rigorous studies since 1956, when Hofstadter introduced the electron scattering techniques[1]. To this day, new and surprising features are being discovered[2–7] and debated[8–10].

A common strategy to achieve a deeper understanding of these features is to investigate the impact of introducing heavy and unstable quarks into the bound system. The lightest siblings of the proton are the $\Lambda$ and the $\Sigma^0$ hyperons, both consisting of an up-quark ($u$), a down-quark ($d$) and a heavy and unstable strange-quark ($s$), in contrast to the proton with a $uud$ structure of only light quarks. Since hyperons are unstable, they cannot be studied in conventional electron scattering experiments ($e^-Y \to e^-Y$, where $Y$ represents the hyperon)[9], which require stable beams or targets. Hyperon-antihyperon annihilation processes (such as $Y\bar{Y} \to \eta e^+ e^-$) are even more challenging and do not constitute a realistic alternative. Instead, time-like form factors of hyperons can be accessed in electron-positron annihilations with the subsequent production of a hyperon-antihyperon pair, such as $e^+ e^- \to \bar{\Lambda}\Sigma^0$. In this scenario, hyperon and antihyperon are quantum spin correlated with same or opposite helicity states for spin-1/2

hyperons, which signifies that the transition from the initial electron-positron pair to the final baryon-antibaryon pair involves amplitudes for both helicity conservation and helicity flip[11]. If there is a non-vanishing phase between the transition amplitudes for these different helicity states, we can observe the polarization of baryons through the angular distribution of the final-state particles. In light of this, the modulus and phase of the ratio $G_E/G_M$ in time-like region can be accessed directly from the measurement of the polarization of one of the outgoing baryons along the direction orthogonal to the scattering plane. The time-like form factors can be seen as snapshots of the time evolution of a hyperon-antihyperon pair. In particular, the modulus and phase of the ratio $G_E/G_M$ in the time-like region are very sensitive to the specifics of the hyperon interaction. Therefore, by dispersive calculations we can constrain the form factors also in the space-like region, gaining profound insight into the inner structure[9,12–14]. The dispersive relation has demonstrated an unprecedented capability to ascertain the intricate nature of the ratio based on its modulus and phase measured at the BESIII Collaboration at a single energy point[9,15]. However, the absence of data makes the predictions quite uncertain. In addition, the asymptotic behaviour of the form factor phase is of special interest at large energies, where the time-like and the space-like form factors should converge to the same real value. Hence, there should be a scale at which the phase approaches an integer multiple of $\pi$. Therefore, gathering additional data at different energy points would be essential to bolster the predictive capacity of the dispersive relation and to reveal additional remarkable attributes of baryons. Precise data at a relatively high energy would therefore be a pivotal step forward in the understanding of dynamics underlying the interaction of hyperons. Especially the $\bar{\Lambda}\Sigma^0$ ($\Lambda\bar{\Sigma}^0$) transition, it is particularly interesting since it is the only ground-state transition for which we can gather data both in the high-energy time-like region (this work) and in the very low-energy region (via Dalitz decays, i.e. $\Sigma^0 \to \Lambda e^+ e^-$)[9]. The prospect of in the future comparing these two different energy regions is therefore unique.

In recent years, the BESIII collaboration has performed pioneering studies of hyperon form factors[16]. In particular, the self-analyzing hyperon decays can be used to measure the hyperon polarization, thereby completely determining the form factors of the $\Lambda$ hyperon[15]. However, time-like form factors need to be studied in processes where a one-photon exchange is the dominating process, as shown in Fig. 1d. For a hyperon-antihyperon pair of the same type, e.g. $\Lambda\bar{\Lambda}$, this means that the electron-positron annihilation must occur at an energy far from any vector meson resonances that can decay strongly into a hyperon-antihyperon pair. For a pair where the hyperon and the anti-hyperon from $J/\psi$ are of different type, e.g. $\Lambda\bar{\Sigma}^0$ or $\bar{\Lambda}\Sigma^0$, since the process is isospin-violating, the purely strong amplitude is suppressed by the small dimensionless factor $\frac{m_d - m_u}{m_c} \sim \frac{1}{500}$, where the $m_u$, $m_d$ and $m_c$ represent the mass of $u$ quark, $d$ quark and $c$ quark, respectively. Therefore, the suppressed strong process involving an intermediate $ggg$ state from the $J/\psi$ decay (Fig. 1a) with a branching fraction of 64.1% according to the Particle Data Group (PDG)[17] is negligible compared to $\gamma gg$ (8.8%) (Fig. 1(b)) and $\gamma^*$ (13.5%) (Fig. 1c) mediated decays. Furthermore, the agreement between the expected coupling to the $J/\psi$ decay and the value extracted from cross section data in the electromagnetic continuum[18], indicates a clear absence of the $\gamma gg$ process in

the $J/\psi \to \bar{\Lambda}\Sigma^0 + c.c.$. Hence, $e^+ e^- \to J/\psi \to \bar{\Lambda}\Sigma^0$ must be a purely electromagnetic process mediated by $\gamma^* \to c\bar{c}(loop) \to \gamma^*$, namely the hadronic vacuum polarization effect, as depicted in Fig. 1c, which has the same final production $\gamma^*\bar{\Lambda}\Sigma^0$ vertex as Fig. 1d. Accordingly, the electric and magnetic form factors of Fig. 1d can be extracted from Fig. 1c by correcting for the well-known vacuum polarization, which exhibits a notable enhancement attributed to the $J/\psi$ resonance.

In this work, using the available $(10087 \pm 44) \times 10^6\ J/\psi$ events produced in $e^+ e^-$ annihilations[19] at BESIII, almost one order of magnitude larger than the data sample used in the previous measurement[11], we investigate the form factors in the reaction $e^+ e^- \to J/\psi \to \bar{\Lambda}\Sigma^0$ with the polarized and spin correlated $\Lambda\bar{\Sigma}^0$ pairs, baryons and anti-baryons simultaneously produced with correlated spins as defined in refs. [20,21]. With the hadronic vacuum polarization at the $J/\psi$ resonance resulting in a significantly enhanced signal, we probe the same vertex as the one-photon exchange process and attain the structure at the $J/\psi$ resonance. The inclusion of charge-conjugate processes is implied hereafter unless explicitly mentioned otherwise.

## Results and discussion
### BESIII detector and candidates selection
The BESIII detector[22] records symmetric $e^+ e^-$ collisions provided by the BEPCII storage ring[23], which operates with a peak luminosity of $10^{33}\ \text{cm}^{-2}\text{s}^{-1}$ in the centre-of-mass energy ($\sqrt{s}$) range from 2.0 to 4.95 GeV. In this cylindrical system, tracks of charged particles in the detector are reconstructed from track-induced signals and the momenta are determined from the track curvature in the main drift chamber (MDC). The flight time of charged particles is recorded by a plastic scintillator time-of-flight system (TOF). Showers from photon clusters are reconstructed and energy deposits are measured in the electromagnetic calorimeter (EMC). The signal of $e^+ e^- \to J/\psi \to \bar{\Lambda}(\to \bar{p}\pi^+)\Sigma^0(\to \gamma\Lambda \to \gamma p\pi^-)$ is extracted from $(10087 \pm 44) \times 10^6\ J/\psi$ events[19] at $\sqrt{s} = 3.097$ GeV, equivalent to an integrated luminosity of 3083 pb$^{-1}$ [19]. The $\Lambda$ ($\bar{\Lambda}$) is reconstructed using $p\pi^-$ ($\bar{p}\pi^+$) decays and $\Sigma^0$ from $\gamma\Lambda$ decays. The specific requirements of event reconstruction and selection criteria are described in the Methods below. The resulting signals of $\bar{\Lambda}(\Lambda)$ and $\Sigma^0(\bar{\Sigma}^0)$ are clearly observed, as shown in Supplementary Figs. 1 and 2. The possible background events are investigated with an inclusive Monte Carlo (MC) sample generated with all known $J/\psi$ decays. To estimate the number of background events coming directly from the continuum light hadron (QED) process, the same analysis is performed on the data sample at $\sqrt{s} = 3.080$ GeV, corresponding to an integrated luminosity of 166.3 pb$^{-1}$ [19]. With an extended unbinned maximum likelihood fit to the $\gamma\Lambda$ ($\gamma\bar{\Lambda}$) invariant mass distribution shown in Supplementary Fig. 3, the final signal yields are determined to be $26260 \pm 181$ and the QED background are $39 \pm 7$. The details of backgrounds analysis and fit are described in the Methods.

### The vacuum polarisation effect in $e^+ e^- \to J/\psi \to \bar{\Lambda}\Sigma^0$
Based on the studies of $e^+ e^- \to \mu^+ \mu^-$ and $\eta\pi^+\pi^-$ in ref. [24] the relative phase between the hadronic vacuum (Fig. 1c) and the continuum (Fig. 1d) processes is zero in case of a purely electromagnetic decay, and it has a line shape similar to the cross section of the purely electromagnetic process. Consequently, the ratio of the cross section at the $J/\psi$ peak to that at any specific energy is the same for different

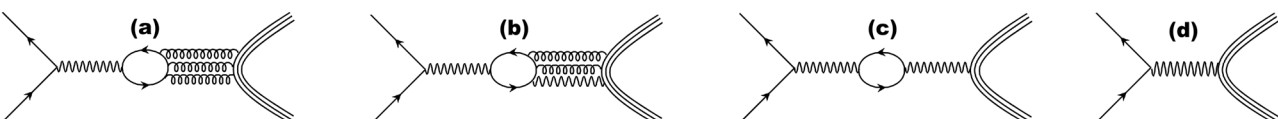

**Fig. 1 | The Feynman diagrams for $e^+ e^- \to$ *hadrons* in the vicinity of the $J/\psi$.** **a** strong process with intermediate $J/\psi$ mediated by gluons ($ggg$), (**b**) the mixed strong-electromagnetic process of $J/\psi$ decay mediated by $\gamma gg$, (**c**) electromagnetic process through the vacuum polarization of one virtual photon ($\gamma^*$) to $J/\psi$, (**d**) continuum process without the $J/\psi$ intermediate state but only one virtual photon.

purely electromagnetic processes as illustrated by both $e^+e^- \to \mu^+\mu^-$ and $\eta\pi^+\pi^-$. With the measured cross sections in ref. 24 the corresponding ratios of these two processes are calculated to be $24.20 \pm 0.81$ and $28.81 \pm 8.52$, respectively, both in good agreement with each other. Here, the uncertainties are statistical only since the systematic uncertainties cancel in the calculation of the ratio. We also performed a measurement of the cross sections of $e^+e^- \to \bar{\Lambda}\Sigma^0 + c.c.$ at the $J/\psi$ peak and 3.08 GeV, determining the corresponding ratio to be $33.72 \pm 6.06$. This value is consistent with those from the above processes within the uncertainties, thus providing further evidence for $J/\psi \to \bar{\Lambda}\Sigma^0 + c.c.$ as a purely electromagnetic decay, which implies a way to extract the electromagnetic form factor with the hadronic vacuum polarization at the $J/\psi$ peak.

Since the imaginary part of form factors is non-zero at centre-of-mass energies above the two-pion threshold[12,25], the relative phase $\Delta\Phi$ between the electric and magnetic form factors, $G_E$ and $G_M$, is expected to be non-zero. In the case of $e^+e^- \to J/\psi \to \bar{\Lambda}\Sigma^0$, a non-vanishing $\Delta\Phi$ also demonstrates the polarization of $\Lambda$ and $\bar{\Sigma}^0$ in the direction perpendicular to the production plane. Since the electron mass is negligible in comparison to the $J/\psi$ mass, the initial electron and positron helicities have to be the opposite. This implies that the angular distribution and polarization can be described uniquely by only two quantities, the relative phase $\Delta\Phi = \arg(G_E/G_M)$ and the angular distribution parameter $\alpha = \frac{s - 4M_Y^2 R^2}{s + 4M_Y^2 R^2}$[26], where $R = |\frac{G_E}{G_M}|$ and $M_Y$ is the mass of the final hyperon. For $\bar{\Lambda}\Sigma^0$ ($\Lambda\bar{\Sigma}^0$), $M_Y$ is replaced by $(M_{\Sigma^0} + M_\Lambda)/2$[27]. The feasibility of extracting the form factors in the production and cascade decays of $e^+e^- \to J/\psi \to \bar{\Lambda}(\to \bar{p}\pi^+)\Sigma^0(\to \gamma\Lambda \to \gamma p\pi^-)$ is described by the six kinematic variables as described in Methods, expressed as the helicity angles $\boldsymbol{\xi} = (\theta, \theta_\Lambda, \phi_\Lambda, \theta_p, \theta_{\bar{p}}, \phi_{\bar{p}})$ shown in Fig. 2.

Here, we denote the angular distribution parameter, the relative phase and decay asymmetries for $\Sigma^0 \to \gamma\Lambda$, $\Lambda \to p\pi^-$, and $\bar{\Lambda} \to \bar{p}\pi^+$ as $\alpha_{J/\psi}$, $\Delta\Phi$, $\alpha_\gamma$, $\alpha_\Lambda$, and $\alpha_{\bar{\Lambda}}$, respectively. Subsequently, to extract the form factors, the helicity analysis is performed for $J/\psi \to \bar{\Lambda}\Sigma^0 + c.c.$ based on the angular distribution as described in detail in the Methods. Although $e^+e^- \to J/\psi \to \bar{\Lambda}\Sigma^0$ and $e^+e^- \to J/\psi \to \Lambda\bar{\Sigma}^0$ are two independent reactions, their helicity amplitudes are simply related before and after charge-conjugate and parity transformation. In accordance with the Standard Model (SM), CP violation is absent in

electromagnetic processes. As a result, the relative phases $\Delta\Phi$ of these two decays are expected to satisfy $\Delta\Phi_{\bar{\Lambda}\Sigma^0} + \Delta\Phi_{\Lambda\bar{\Sigma}^0} = \pi$, where $\Delta\Phi_{\bar{\Lambda}\Sigma^0}$ and $\Delta\Phi_{\Lambda\bar{\Sigma}^0}$ denote the relative phases of time-like electric and magnetic form factors for $e^+e^- \to J/\psi \to \bar{\Lambda}\Sigma^0$ and $e^+e^- \to J/\psi \to \Lambda\bar{\Sigma}^0$, respectively. Therefore, a simultaneous measurement of $\bar{\Lambda}\Sigma^0$ and $\Lambda\bar{\Sigma}^0$ offers the possibility of exploring CP violation by evaluating $\Delta\Phi_{CP} = |\pi - (\Delta\Phi_{\bar{\Lambda}\Sigma^0} + \Delta\Phi_{\Lambda\bar{\Sigma}^0})|$, which is required to be zero from CP invariance within the SM. In this case, these processes are also of interest for searching for additional sources of CP violation beyond the SM.

In the $\Sigma$ mass region, a combined helicity analysis is performed for $J/\psi \to \bar{\Lambda}\Sigma^0$ and $J/\psi \to \Lambda\bar{\Sigma}^0$ and the parameters $\alpha_\Lambda$ and $\alpha_{\bar{\Lambda}}$ are fixed to be $\alpha_\Lambda = 0.7519$ and $\alpha_{\bar{\Lambda}} = -0.7559$[28] from previous high-precision measurements of $J/\psi \to \Lambda\bar{\Lambda}$. Using the average magnitude for both has a negligible effect on fit results. Due to the electromagnetic part of the decay chain, $\Sigma^0 \to \gamma\Lambda$, where the photon polarization is not measured[29], the $\alpha_\gamma$ is presumed to be 0. The free parameters, including $\alpha_{J/\psi}$ and the relative phase $\Delta\Phi_{\bar{\Lambda}\Sigma^0}$ ($\Delta\Phi_{\Lambda\bar{\Sigma}^0}$) for $e^+e^- \to J/\psi \to \bar{\Lambda}\Sigma^0$ ($\Lambda\bar{\Sigma}^0$), are optimized with an unbinned maximum likelihood fit defined in Methods. These parameters are measured by incorporating the transverse polarization of $\Sigma^0(\bar{\Sigma}^0)$ in the joint angular distribution. The global fit is represented by the multidimensional angular distributions shown in Supplementary Figs. 4 and 5 with a specific fitting technique as well as systematic uncertainties described in Methods.

### Extraction of the form factor ratio and test of the CP violation
From the global fit, a prominent polarization and strong correlation of the relative phase between the two processes are observed, characterized by $P_y$ elucidating the spin transverse polarization and $C_{xz}$ representing the particular relationship between $\Delta\Phi_{\bar{\Lambda}\Sigma^0}$ and $\Delta\Phi_{\Lambda\bar{\Sigma}^0}$.

Their strong dependence on the $\Sigma^0$ ($\bar{\Sigma}^0$) direction angle $\theta$, defined in the Methods, is seen in Fig. 3. To illustrate the fit quality, the fit results in each $\cos\theta_{\Sigma^0/\bar{\Sigma}^0}$ bin are also shown using points with error bars in Fig. 3. Apart from the difference caused by the fluctuations from the complex background channels, the points of each bin are consistent with the globally fitted curves. The fit yields $\alpha_{J/\psi} = 0.418 \pm 0.028(\text{stat.}) \pm 0.014(\text{syst.})$, $\Delta\Phi_{\bar{\Lambda}\Sigma^0} = (1.011 \pm 0.094(\text{stat.}) \pm 0.010(\text{syst.}))\,rad$, and $\Delta\Phi_{\Lambda\bar{\Sigma}^0} = (2.128 \pm 0.094(\text{stat.}) \pm 0.010(\text{syst.}))\,rad$.

The ratio $R = |\frac{G_E}{G_M}| = \frac{\sqrt{s}}{2M_Y}\sqrt{\frac{1-\alpha}{1+\alpha}}$ is determined to be $0.860 \pm$

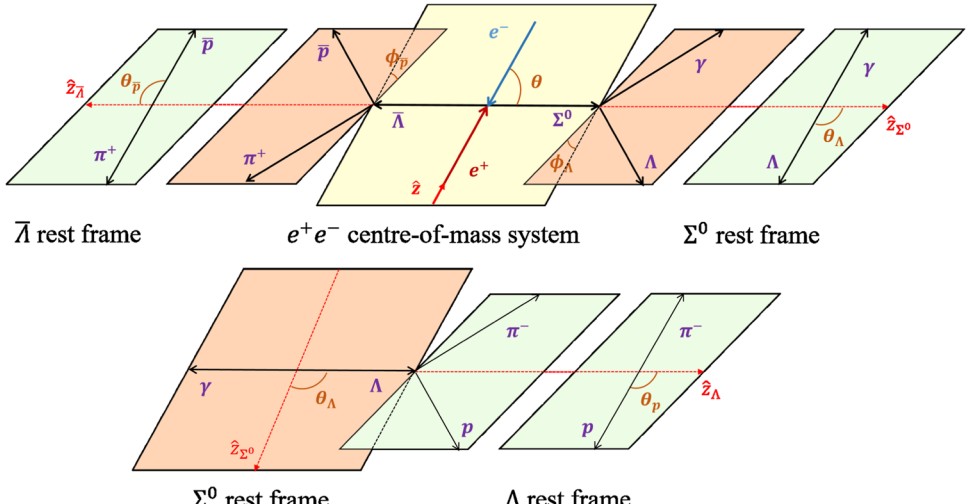

**Fig. 2 | Definition of the helicity angles for $J/\psi \to \bar{\Lambda}(\to \bar{p}\pi^+)\Sigma^0(\to \gamma\Lambda \to \gamma p\pi^-)$.** The angles $\theta$, $\theta_\Lambda$, $\theta_p$, $\theta_{\bar{p}}$ are the polar helicity angles of the $\Sigma^0$, $\Lambda$, $p$ and $\bar{p}$ in the $e^+e^-$ centre-of-mass system, $\Sigma^0$ rest frame, $\Lambda$ rest frame and $\bar{\Lambda}$ rest frame, respectively. The angles between different decay or production planes, $\phi_\Lambda$ and $\phi_{\bar{p}}$, are the azimuthal helicity angles of the $\Lambda$ and $\bar{p}$ in the $\Sigma^0$ rest frame and $\Lambda$ rest frame,

respectively. In the $e^+e^-$ centre-of-mass system, the $\mathbf{z}$ is along the $e^+$ momentum direction, and the $\mathbf{z}_\Sigma$ is along the $\Sigma^0$ outgoing direction. In the $\Sigma^0$ rest frame, the polar axis is $\mathbf{z}_\Sigma$, $\mathbf{y}_\Sigma$ is along $\mathbf{z} \times \mathbf{z}_\Sigma$ and $\mathbf{z}_\Lambda$ is along the $\Lambda$ outgoing direction. In the $\Lambda$ rest frame, the polar axis is $\mathbf{z}_\Lambda$, and $\mathbf{y}_\Lambda$ is along $\mathbf{z}_\Sigma \times \mathbf{z}_\Lambda$. In the $\bar{\Lambda}$ rest frame, the polar axis is $\mathbf{z}_{\bar{\Lambda}}$, and $\mathbf{y}_{\bar{\Lambda}}$ is along $\mathbf{z} \times \mathbf{z}_{\bar{\Lambda}}$.

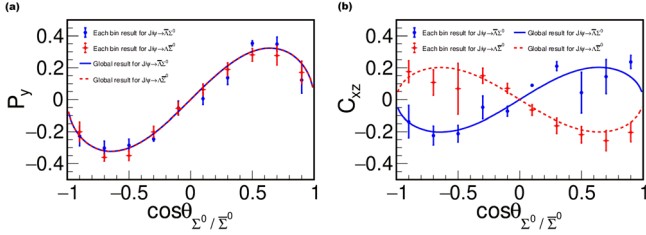

**Fig. 3 | Polarization in and spin correlations of the $e^+e^- \to J/\psi \to \bar{\Lambda}\Sigma^0 \sim (\Lambda\bar{\Sigma}^0)$ reaction.** The points with error bars, blue solid dot for $J/\psi \to \bar{\Lambda}\Sigma^0$ and red open double diamond for $J/\psi \to \Lambda\bar{\Sigma}^0$, are extracted in each $\cos\theta_{\Sigma^0}$ ($\cos\theta_{\bar{\Sigma}^0}$) bin, and the blue solid curves denote the global expected dependence on $\cos\theta_{\Sigma^0}$ ($\cos\theta_{\bar{\Sigma}^0}$ for the red dotted curve).

0.029(stat.) $\pm$ 0.015(syst.), giving the ratio and relative phase of the electric and magnetic form factors $G_E$ and $G_M$ for $e^+e^- \to J/\psi \to \bar{\Lambda}\Sigma^0$ ($\Lambda\bar{\Sigma}^0$) at $\sqrt{s} = 3.097$ GeV, with clear transverse spin polarizations of the $\Lambda$ and $\bar{\Sigma}^0$ observed. The sum of these two relative phases, $\Delta\Phi_{\bar{\Lambda}\Sigma^0} + \Delta\Phi_{\Lambda\bar{\Sigma}^0} = (3.139 \pm 0.133(\text{stat.}) \pm 0.014(\text{syst.}))\,rad$, is in good agreement with the expected value of $\pi$. $\Delta\Phi_{CP} = |\pi - (\Delta\Phi_{\bar{\Lambda}\Sigma^0} + \Delta\Phi_{\Lambda\bar{\Sigma}^0})|$ is calculated to be $0.003 \pm 0.133(\text{stat.}) \pm 0.014(\text{syst.})$, which is consistent with zero and indicates no evident direct CP violation in the decays of $J/\psi \to \bar{\Lambda}\Sigma^0$ and $J/\psi \to \Lambda\bar{\Sigma}^0$. This is the measurement that the time-like structure for $e^+e^- \to \bar{\Lambda}\Sigma^0 + c.c.$ is extracted at $\sqrt{s} = 3.097$ GeV with high precision by using the hadronic vacuum polarization enhancement at the $J/\psi$. In addition, unlike $e^+e^-$ annihilation into hyperon anti-hyperon pairs, $\Lambda$ and $\bar{\Sigma}^0$ are not charge conjugates of each other, which enables us to explore direct CP violation by comparison of polarizations from both $e^+e^- \to J/\psi \to \Lambda\bar{\Sigma}^0$ and $e^+e^- \to J/\psi \to \bar{\Lambda}\Sigma^0$. While currently statistically limited, it provides a way to search for possible new sources of CP violation. In the future, the BESIII experiment may provide even greater sensitivity to direct CP violation[30], with further improvement expected from the next generation experiments, e.g., the next-generation tau-charm physics facility[31] and PANDA[32].

## Methods
### Monte Carlo simulation
The optimization of the event selection criteria and the estimation of physics background as well as the determination of efficiency are performed using MC simulated samples. The GEANT4-based[33] MC package includes the geometric description of the BESIII detector and the detector response. The inclusive MC sample includes both the production of the $J/\psi$ resonance and the continuum processes incorporated in KKMC[34]. All particle decays are modelled with EVTGEN[35,36] using branching fractions either taken from the Particle Data Group (PDG)[17], when available, or otherwise estimated with LUNDCHARM[37,38]. For the signal $J/\psi \to \bar{\Lambda}\Sigma^0 + c.c.$, the MC samples are produced using the angular distribution formula shown in the Methods of Helicity amplitudes. For the determination of the cross section, the generator CONEXC[39] was used. For the background channels $J/\psi \to \Sigma^0\bar{\Sigma}^0$, $J/\psi \to \Lambda\bar{\Lambda}$, the exclusive MC samples were generated in accordance with their decay amplitudes[11,40].

### Initial selection criteria
Candidates for $J/\psi \to \bar{\Lambda}(\to \bar{p}\pi^+)\Sigma^0(\to \gamma\Lambda \to \gamma p\pi^-)$ are required to have four charged tracks with net zero charge and at least one photon.

Charged tracks are selected in the MDC within $\pm 20$ cm of the interaction point in the beam direction and within 10 cm in the plane perpendicular to the beam. The polar angles of these tracks are required to be within the MDC fiducial volume, $|\cos\theta| < 0.93$, where $\theta$ is defined with respect to the z-axis, which is the symmetry axis of the MDC. No particle identification is used to maintain high efficiency.

To reconstruct the decays $\Lambda \to p\pi^-$ and $\bar{\Lambda} \to \bar{p}\pi^+$, we loop over all the combinations of positive and negative charged track pairs and require that at least one $(p\pi^-)(\bar{p}\pi^+)$ track hypothesis successfully passes the vertex finding algorithm[41] of $\Lambda$ and $\bar{\Lambda}$. If more than one accepted combination satisfies the vertex fit requirement, the one with the minimum value of $\sqrt{(M_{p\pi^-} - M_\Lambda)^2 + (M_{\bar{p}\pi^+} - M_\Lambda)^2}$ is chosen, where $M_{p\pi^-}(M_{\bar{p}\pi^+})$ is the $p\pi^-$ ($\bar{p}\pi^+$) invariant mass and $M_\Lambda$ is the nominal $\Lambda$ mass[17].

For good photon selection, showers in the EMC identified as photon candidates are required to satisfy fiducial and shower-quality requirements. For the barrel region, showers must have a minimum energy deposition of 25 MeV with the polar angle of each track satisfying $|\cos\theta| < 0.80$, while those from the end cap region must have at least 50 MeV and the polar angle is required to be $0.86 < |\cos\theta| < 0.92$. To suppress background noise unrelated to the event, the difference between the EMC time and the event start time (TDC) has to fulfil $0 \le TDC \le 700$ ns. To suppress showers generated by charged particles, the photon candidate angular separation from the nearest charged track is required to be at least 10°.

The selected events are subjected to a four-constraint energy momentum conservation kinematic fit (4C fit) with the hypothesis of $\gamma\Lambda\bar{\Lambda}$. The kinematic fit adjusts the reconstructed particle energy and momentum within the measured errors so as to satisfy energy and momentum conservation for the given event hypothesis. This improves resolution and reduces background. When there are multiple photon candidates in an event, the combination with the smallest $\chi^2_{4C}$ is retained. The kinematic fit is very powerful to suppress background events with multiple photon candidates in the final states, e.g., $J/\psi \to \Sigma^0\bar{\Sigma}^0$ and $J/\psi \to \Lambda\bar{\Sigma}^0\pi^0$.

### Final selection criteria
After the initial selection, the scatter plot of $M_{p\pi^-}$ versus $M_{\bar{p}\pi^+}$ of the accepted candidates is shown in Supplementary Fig. 1, where the clear cluster corresponds to the decays of $\Lambda \to p\pi^-$ and $\bar{\Lambda} \to \bar{p}\pi^+$. The $\Lambda$ and $\bar{\Lambda}$ signal candidates are selected by requiring $|M_{p\pi^-} - M_\Lambda| < 5$ MeV/$c^2$ and $|M_{\bar{p}\pi^+} - M_\Lambda| < 5$ MeV/$c^2$. To further suppress backgrounds and improve the mass resolution, the 4C kinematic fit must satisfy $\chi^2_{4C} < 30$. In addition, $M_{\gamma\bar{\Lambda}} > 1.135$ GeV/$c^2$ and $M_{\gamma\Lambda} > 1.135$ GeV/$c^2$ are required in the further analysis for $J/\psi \to \bar{\Lambda}\Sigma^0$ and $J/\psi \to \Lambda\bar{\Sigma}^0$, respectively, which has a pronounced effect on suppressing the background events from $J/\psi \to \Lambda\bar{\Lambda}$. After applying the above requirements, the invariant mass spectrum of $\gamma\Lambda$ ($\gamma\bar{\Lambda}$) is shown in Supplementary Fig. 2, where the prominent peak of $\Sigma^0$ ($\bar{\Sigma}^0$) is clearly observed.

### Background analysis
Possible background sources are investigated with an inclusive MC sample of 10 billion $J/\psi$ decays. Using the same selection criteria, with the help of a generic event type analysis tool[42], the surviving background events mainly originate from $J/\psi \to \Sigma^0\bar{\Sigma}^0$, $J/\psi \to \Lambda\bar{\Lambda}$ and $J/\psi \to \gamma\Lambda\bar{\Lambda}$ (including a resonant contribution from $\gamma\eta_c$), but none of these produce an evident peak in the $\Sigma^0$ mass region. The exclusive MC samples of these background channels are generated with the corresponding helicity amplitudes and their contributions are shown in Supplementary Fig. 2. To estimate the number of background events coming directly from the $e^+e^-$ annihilation, the same analysis is performed on data taken at $\sqrt{s} = 3.080$ GeV, where the number of background events, $39 \pm 7$ is also extracted by fitting the $\gamma\Lambda$ (or $\gamma\bar{\Lambda}$) mass spectrum as shown in Supplementary Fig. 3. The background events are then normalized to the $J/\psi$ data after taking into account the

luminosities and energy-dependent cross sections of continuum processes[43], with the scaling factor calculated as

$$f = \frac{\mathcal{L}_{J/\psi}}{\mathcal{L}_{\psi(3080)}} \times \frac{s^5_{\psi(3080)}}{s^5_{J/\psi}} \times \frac{\epsilon_{\psi(3080)}}{\epsilon_{J/\psi}}. \tag{1}$$

Here, $\mathcal{L}$, $s$, and $\epsilon$ are the integrated luminosity, the square of the centre-of-mass energy, and the detection efficiency at the two centre-of-mass energies, respectively. the number of background events for $J/\psi \to \bar{\Lambda}\Sigma^0$ is normalized to be $669 \pm 120$. It should be pointed out that there is no interference between the QED background and the $J/\psi$ resonance since this is a purely electromagnetic process according to ref. 18.

## Signal extraction

The signal yields are obtained from an extended unbinned maximum likelihood fit to the $\gamma\Lambda$ ($\gamma\bar{\Lambda}$) mass spectrum. The total probability density function (PDF) consists of a signal and various background contributions. The signal component is modelled as the MC simulated signal shape convolved with a Gaussian function to account for the difference in the mass resolution between data and MC simulation. The background components, $J/\psi \to \Sigma^0\bar{\Sigma}^0$, $J/\psi \to \Lambda\bar{\Lambda}$, and $J/\psi \to \gamma\Lambda\bar{\Lambda}$ ($\gamma\eta_c$), as well as the reflection from signal conjugation decay mode, are described with the simulated shapes derived from the dedicated MC samples, while the magnitudes of different components are left free to account for the uncertainties of the branching fractions of these decays and other intermediate decays. The fit to the $M_{\gamma\Lambda}/M_{\gamma\bar{\Lambda}}$ spectrum, as displayed in Supplementary Fig. 2, gives $26260 \pm 181$ $\bar{\Lambda}\Sigma^0$ events.

## Helicity amplitude

The structure of the six dimensional angular distribution is determined by global parameters $\boldsymbol{\omega} = (\alpha_{J/\psi}, \Delta\Phi, \alpha_\gamma, \alpha_\Lambda, \alpha_{\bar{\Lambda}})$ independent of the $\Sigma^0$ scattering angle, $\theta_{\Sigma^0}$, and is written in a modular form as

$$\mathcal{W}(\boldsymbol{\xi}; \boldsymbol{\omega}) = \sum_{\mu,\nu=0}^{3} \sum_{\mu'=0}^{3} C_{\mu\nu} a^{\Sigma^0}_{\mu\mu'} a^\Lambda_{\mu'0} a^{\bar{\Lambda}}_{\nu 0}, \tag{2}$$

where the $C_{\mu\nu}(\theta; \alpha_{J/\psi}, \Delta\Phi)$ is a $4 \times 4$ spin density matrix, describing the spin configuration of the spin correlated hyperon-antihyperon pair. The matrix elements are expressed as

$$C_{\mu\nu} = (1 + \alpha_{J/\psi} \cos^2\theta) \begin{pmatrix} 1 & 0 & P_y & 0 \\ 0 & C_{xx} & 0 & C_{xz} \\ -P_y & 0 & C_{yy} & 0 \\ 0 & -C_{xz} & 0 & C_{zz} \end{pmatrix}, \tag{3}$$

where $P_y$ governs the polarization of the $\Sigma^0$ and $C_{ij}$ characterizes its spin correlations. Both $P_y$ and $C_{ij}$ can be written in terms of $\sin\Delta\Phi$ or $\cos\Delta\Phi$ as

$$P_y = f(\theta)\sin\Delta\Phi, C_{xz} = f(\theta)\cos\Delta\Phi, \tag{4}$$

where $f(\theta)$, a common function dependent on the $\Sigma^0$ ($\bar{\Sigma}^0$) direction angle $\theta$, is expressed as

$$f(\theta) = \frac{\sqrt{1 - \alpha^2_{J/\psi}}\sin\theta\cos\theta}{1 + \alpha_{J/\psi}\cos^2\theta}. \tag{5}$$

The matrices $a^\gamma_{\mu\nu}$ in Eq. (2) represent the propagation of the spin density matrices in the sequential decays. The full expressions for $C_{\mu\nu}$ and $a^\gamma_{\mu\nu}$ are given in refs. 44,38.

## Global fit of parameters

A non-zero phase angle difference $\Delta\Phi$ indicates transverse hyperon polarization, which allows us to measure these parameters at the same time. A simultaneous fit is performed to the two conjugate channels, $J/\psi \to \bar{\Lambda}\Sigma^0$ and $J/\psi \to \Lambda\bar{\Sigma}^0$. The likelihood function constructed from the probability density function for an event characterized by $\boldsymbol{\xi}_i$ is

$$\mathcal{L} = \prod_{i=1}^{N} \mathcal{P}(\boldsymbol{\xi}_i; \boldsymbol{\omega}) = \prod_{i=1}^{N} \frac{\mathcal{W}(\boldsymbol{\xi}_i; \boldsymbol{\omega})\epsilon(\boldsymbol{\xi}_i)}{\mathcal{N}(\boldsymbol{\omega})}, \tag{6}$$

where $\epsilon(\boldsymbol{\xi}_i)$ is the detection efficiency, $N$ is the number of the surviving data events after all selection criteria, the normalization factor $\mathcal{N}(\boldsymbol{\omega}) = \int \mathcal{W}(\boldsymbol{\xi}; \boldsymbol{\omega})\epsilon(\boldsymbol{\xi})d\boldsymbol{\xi}$, with $\mathcal{W}(\boldsymbol{\xi}; \boldsymbol{\omega})$ defined in Eq. (2), and $\mathcal{P}$ is the probability to produce event $i$ based on the measured parameters $\boldsymbol{\xi}_i$ and the set of observables $\boldsymbol{\omega}$. Based on the likelihood function defined in Eq. (6), the objective function is written as

$$S = -\ln\mathcal{L}^I_{data} - \ln\mathcal{L}^{II}_{data} + \ln\mathcal{L}^I_{bkg} + \ln\mathcal{L}^{II}_{bkg}, \tag{7}$$

where $\ln\mathcal{L}^{I,II}_{data}$ and $\ln\mathcal{L}^{I,II}_{bkg}$ are the likelihood functions for $J/\psi \to \bar{\Lambda}\Sigma^0$ and $J/\psi \to \Lambda\bar{\Sigma}^0$ and the background events from simulation, respectively. In order to optimize the free parameters ($\alpha_{J/\psi}$, $\Delta\Phi_{\bar{\Lambda}\Sigma^0}$ and $\Delta\Phi_{\Lambda\bar{\Sigma}^0}$) and minimize the objective function, the normalization factor $\mathcal{N}(\boldsymbol{\omega})$ in Eq. (6) is obtained by MC integral generated by phase space through all event selection criteria. We adjust the weights of the phase space sample events to match the momentum distribution of the final-state particles to the data. The weighted phase space events can then be employed to construct distributions of various physical quantities, thus displaying the fit results. To compare the fit with data, the moments directly related to helicity amplitude are defined as:

$$\begin{aligned} T_1 &= \sum_{i}^{N_k} \left( \cos^2\theta\, n^{(i)}_{1,z} n^{(i)}_{2,z} - \sin^2\theta\, n^{(i)}_{1,x} n^{(i)}_{2,x} \right), \\ T_2 &= \sum_{i}^{N_k} \cos\theta\sin\theta \left( n^{(i)}_{1,z} n^{(i)}_{2,x} - n^{(i)}_{1,x} n^{(i)}_{2,z} \right), \\ T_3 &= \sum_{i}^{N_k} \cos\theta\sin\theta\, n^{(i)}_{1,y}, \\ T_4 &= \sum_{i}^{N_k} \cos\theta\sin\theta\, n^{(i)}_{2,y}, \\ T_5 &= \sum_{i}^{N_k} \left( n^{(i)}_{1,z} n^{(i)}_{2,z} - \sin^2\theta\, n^{(i)}_{1,y} n^{(i)}_{2,y} \right), \end{aligned} \tag{8}$$

where $N_k$ is the number of events in the $k^{th}$ $\cos\theta$ bin and $\mathbf{n}_1(\mathbf{n}_2)$ is the unit vector in the direction of the nucleon (anti-nucleon) in the rest frame of $\Sigma^0$ ($\bar{\Lambda}$) for $J/\psi \to \bar{\Lambda}\Sigma^0$, as illustrated in Fig. 2. The resulting $T_i$ and helicity angle distributions for data and the fit results are shown in Supplementary Figs. 4 and 5, and the difference between $T_3$ and $T_4$ results from the transverse polarization of $\Sigma^0$ ($\bar{\Sigma}^0$), which allows the relative phase between $G_E$ and $G_M$ to be determined from the global fit of polarization with the modulus of the ratio between $G_E$ and $G_M$ obtained from $\alpha = \frac{s - 4M^2_\gamma R^2}{s + 4M^2_\gamma R^2}$.

## Systematic uncertainty

The uncertainties in the measurement of the form factors are mainly from the $\Lambda, \bar{\Lambda}$ reconstruction, the 4C kinematic fit, and the background estimation. For the $\Lambda, \bar{\Lambda}$ reconstruction, a correction to the MC efficiency is made. We also use the control sample of $J/\psi \to \bar{p}K^+\Lambda$ to obtain the efficiencies of the data and MC simulation in the $\Lambda$ and $\bar{\Lambda}$ reconstruction, and then correct the MC efficiencies by the observed data-MC efficiency differences. In order to reduce the impact of statistical fluctuations, the fit with the corrected MC sample is performed

400 times by varying the correction factor randomly within one standard deviation. The differences between the results with and without correction are taken as the systematic uncertainties. For the 4C kinematic fit, the MC sample in the polarization fit is altered by changing the helix parameters of charged tracks, and the same fit procedure is performed to the same data sample. The relative differences of the fit results are assigned as the uncertainties. The systematic uncertainty arising from the background estimate for each background source is assigned by varying the normalization factor by one standard deviation, the maximum change of the result is assigned as the associated systematic uncertainty. The total systematic uncertainty due to the background estimate is obtained by adding all effects of various background sources in quadrature. The uncertainties due to the $\alpha_{\Lambda,\bar{\Lambda}}$ are estimated by varying the quoted value from ref. 28 within one standard deviation. The systematic uncertainties for the polarization measurement, as discussed above, are listed in Supplementary Table 1.

## Data availability
The raw data generated in this study have been deposited in the Institute of High Energy Physics mass storage silo database. The source data are available under restricted access for the complexity and large size, access can be obtained by contacting to besiii-publications@ihep.ac.cn.

## Code availability
All algorithms used for data analysis and simulation are archived by the authors and are available on request to besiii-publications@ihep.ac.cn.

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

## Acknowledgements

The BESIII collaboration thanks the staff of BEPCII and the IHEP computing center for their strong support. This work is supported in part by National Key R&D Program of China under Contracts Nos. 2020YFA0406300, 2020YFA0406400; National Natural Science Foundation of China (NSFC) under Contracts Nos. 11635010, 11735014, 11835012, 11875115, 11935015, 11935016, 11935018, 11961141012, 12022510, 12025502, 12035009, 12035013, 12075250, 12165022, 12192260, 12192261, 12192262, 12192263, 12192264, 12192265, 12225509; the Chinese Academy of Sciences (CAS) Large-Scale Scientific Facility Program; Joint Large-Scale Scientific Facility Funds of the NSFC and CAS under Contract No. U1832207; the CAS Center for Excellence in Particle Physics (CCEPP); 100 Talents Program of CAS; The Institute of Nuclear and Particle Physics (INPAC) and Shanghai Key Laboratory for Particle Physics and Cosmology; Yunnan Fundamental Research Project under Contract No. 202301AT070162; ERC under Contract No. 758462; European Union's Horizon 2020 research and innovation programme under Marie Sklodowska-Curie grant agreement under Contract No. 894790; German Research Foundation DFG under Contracts Nos. 443159800, 455635585, Collaborative Research Center CRC 1044, FOR5327, GRK 2149; Istituto Nazionale di Fisica Nucleare, Italy; Ministry of Development of Turkey under Contract No. DPT2006K-120470; National Science and Technology fund; National Science Research and Innovation Fund (NSRF) via the Program Management Unit for Human Resources & Institutional Development, Research and Innovation under Contract No. B16F640076; Olle Engkvist Foundation under Contract No. 200-0605; STFC (United Kingdom); Suranaree University of Technology (SUT), Thailand Science Research and Innovation (TSRI), and National Science Research and Innovation Fund (NSRF) under Contract No. 160355; Polish National Science Centre under Contract 2019/35/O/ST2/02907; The Royal Society, UK under Contracts Nos. DH140054, DH160214; The Knut and Alice Wallenberg Foundation (Sweden); The Swedish Research Council; The Swedish Foundation for International Cooperation in Research and Higher Education (STINT); U. S. Department of Energy under Contract No. DE-FG02-05ER41374.

## Author contributions

All the authors have contributed to this publication, being variously involved in the design and construction of the detectors, writing software, calibrating sub-systems, operating the detectors, acquiring data and analysing the processed data.

## Competing interests

The authors declare no competing interests.

## Additional information

## The BESIII Collaboration

M. Ablikim[1], M. N. Achasov[2,85], P. Adlarson[3], M. Albrecht[4], R. Aliberti[5], A. Amoroso[6,7], M. R. An[8], Q. An[9,10], Y. Bai[11], O. Bakina[12], R. Baldini Ferroli[13], I. Balossino[14], Y. Ban[15,86], V. Batozskaya[1,16], D. Becker[5], K. Begzsuren[17], N. Berger[5], M. Bertani[13], D. Bettoni[14], F. Bianchi[6,7], E. Bianco[6,7], J. Bloms[18], A. Bortone[6,7], I. Boyko[12], R. A. Briere[19], A. Brueggemann[18], H. Cai[20], X. Cai[1,9], A. Calcaterra[13], G. F. Cao[1,21], N. Cao[1,21], S. A. Cetin[22], J. F. Chang[1,9], W. L. Chang[1,21], G. R. Che[23], G. Chelkov[12,87], C. Chen[23], Chao Chen[24], G. Chen[1], H. S. Chen[1,21], M. L. Chen[1,9,21], S. J. Chen[25], S. M. Chen[26], T. Chen[1,21], X. R. Chen[21,27], X. T. Chen[1,21], Y. B. Chen[1,9], Z. J. Chen[28,88], W. S. Cheng[7], S. K. Choi[24], X. Chu[23], G. Cibinetto[14], F. Cossio[7], J. J. Cui[29], H. L. Dai[1,9], J. P. Dai[30], A. Dbeyssi[31], R. E. de Boer[4], D. Dedovich[12], Z. Y. Deng[1], A. Denig[5], I. Denysenko[12], M. Destefanis[6,7], F. De Mori[6,7], Y. Ding[32], Y. Ding[33], J. Dong[1,9], L. Y. Dong[1,21], M. Y. Dong[1,9,21], X. Dong[20], S. X. Du[34], Z. H. Duan[25], P. Egorov[12,87], Y. L. Fan[20], J. Fang[1,9], S. S. Fang[1,21], W. X. Fang[1], Y. Fang[1], R. Farinelli[14], L. Fava[7,35], F. Feldbauer[4],

G. Felici[13], C. Q. Feng[9,10], J. H. Feng[36], K. Fischer[37], M. Fritsch[4], C. Fritzsch[18], C. D. Fu[1], H. Gao[21], Y. N. Gao[15,86], Yang Gao[9,10], S. Garbolino[7], I. Garzia[14,38], P. T. Ge[20], Z. W. Ge[25], C. Geng[36], E. M. Gersabeck[39], A. Gilman[37], K. Goetzen[40], L. Gong[33], W. X. Gong[1,9], W. Gradl[5], M. Greco[6,7], L. M. Gu[25], M. H. Gu[1,9], Y. T. Gu[41], C. Y. Guan[1,21], A. Q. Guo[21,27], L. B. Guo[42], R. P. Guo[43], Y. P. Guo[44,89], A. Guskov[12,87], W. Y. Han[8], X. Q. Hao[45], F. A. Harris[46], K. K. He[24], K. L. He[1,21], F. H. Heinsius[4], C. H. Heinz[5], Y. K. Heng[1,9,21], C. Herold[47], G. Y. Hou[1,21], Y. R. Hou[21], Z. L. Hou[1], H. M. Hu[1,21], J. F. Hu[48,90], T. Hu[1,9,21], Y. Hu[1], G. S. Huang[9,10], K. X. Huang[36], L. Q. Huang[21,27], X. T. Huang[29], Y. P. Huang[1], Z. Huang[15,86], T. Hussain[49], N. Hüsken[5,50], W. Imoehl[50], M. Irshad[9,10], J. Jackson[50], S. Jaeger[4], S. Janchiv[17], E. Jang[24], J. H. Jeong[24], Q. Ji[1], Q. P. Ji[45], X. B. Ji[1,21], X. L. Ji[1,9], Y. Y. Ji[29], Z. K. Jia[9,10], P. C. Jiang[15,86], S. S. Jiang[8], X. S. Jiang[1,9,21], Y. Jiang[21], J. B. Jiao[29], Z. Jiao[51], S. Jin[25], Y. Jin[52], M. Q. Jing[1,21], T. Johansson[3], S. Kabana[53], N. Kalantar-Nayestanaki[54], X. L. Kang[55], X. S. Kang[33], R. Kappert[54], M. Kavatsyuk[54], B. C. Ke[34], I. K. Keshk[4], A. Khoukaz[18], R. Kiuchi[1], R. Kliemt[40], L. Koch[56], O. B. Kolcu[22], B. Kopf[4], M. Kuemmel[4], M. Kuessner[4], A. Kupsc[3,16], W. Kühn[56], J. J. Lane[39], J. S. Lange[56], P. Larin[31], A. Lavania[57], L. Lavezzi[6,7], T. T. Lei[10,91], Z. H. Lei[9,10], H. Leithoff[5], M. Lellmann[5], T. Lenz[5], C. Li[58], C. Li[23], C. H. Li[8], Cheng Li[9,10], D. M. Li[34], F. Li[1,9], G. Li[1], H. Li[59], H. Li[9,10], H. B. Li[1,21], H. J. Li[45], H. N. Li[48,90], J. Q. Li[4], J. S. Li[36], J. W. Li[29], Ke Li[1], L. J. Li[1,21], L. K. Li[1], Lei Li[60], M. H. Li[23], P. R. Li[61,91,92], S. X. Li[44], S. Y. Li[26], T. Li[29], W. D. Li[1,21], W. G. Li[1], X. H. Li[9,10], X. L. Li[29], Xiaoyu Li[1,21], Y. G. Li[15,86], Z. X. Li[41], Z. Y. Li[36], C. Liang[25], H. Liang[1,21], H. Liang[9,10], H. Liang[32], Y. F. Liang[62], Y. T. Liang[21,27], G. R. Liao[63], L. Z. Liao[29], J. Libby[57], A. Limphirat[47], C. X. Lin[36], D. X. Lin[21,27], T. Lin[1], B. J. Liu[1], C. Liu[32], C. X. Liu[1], D. Liu[10,31], F. H. Liu[64], Fang Liu[1], Feng Liu[65], G. M. Liu[48,90], H. Liu[61,91,92], H. B. Liu[41], H. M. Liu[1,21], Huanhuan Liu[1], Huihui Liu[66], J. B. Liu[9,10], J. L. Liu[67], J. Y. Liu[1,21], K. Liu[1], K. Y. Liu[33], Ke Liu[68], L. Liu[9,10], Lu Liu[23], M. H. Liu[44,89], P. L. Liu[1], Q. Liu[21], S. B. Liu[9,10], T. Liu[44,89], W. K. Liu[23], W. M. Liu[9,10], X. Liu[61,91,92], Y. Liu[61,91,92], Y. B. Liu[23], Z. A. Liu[1,9,21], Z. Q. Liu[29], X. C. Lou[1,9,21], F. X. Lu[36], H. J. Lu[51], J. G. Lu[1,9], X. L. Lu[1], Y. Lu[69], Y. P. Lu[1,9], Z. H. Lu[1,21], C. L. Luo[42], M. X. Luo[70], T. Luo[44,89], X. L. Luo[1,9], X. R. Lyu[21], Y. F. Lyu[23], F. C. Ma[33], H. L. Ma[1], L. L. Ma[29], M. M. Ma[1,21], Q. M. Ma[1], R. Q. Ma[1,21], R. T. Ma[21], X. Y. Ma[1,9], Y. Ma[15,86], F. E. Maas[31], M. Maggiora[6,7], S. Maldaner[4], S. Malde[37], Q. A. Malik[49], A. Mangoni[71], Y. J. Mao[15,86], Z. P. Mao[1], S. Marcello[6,7], Z. X. Meng[52], J. G. Messchendorp[40,54], G. Mezzadri[14], H. Miao[1,21], T. J. Min[25], R. E. Mitchell[50], X. H. Mo[1,9,21], N. Yu Muchnoi[2,85], Y. Nefedov[12], Y. Nerling[31,93], I. B. Nikolaev[2,85], Z. Ning[1,9], S. Nisar[72,94], Y. Niu[29], S. L. Olsen[21], Q. Ouyang[1,9,21], S. Pacetti[71,73], X. Pan[24], Y. Pan[11], A. Pathak[32], Y. P. Pei[9,10], M. Pelizaeus[4], H. P. Peng[9,10], K. Peters[40,93], J. L. Ping[42], R. G. Ping[1,21], S. Plura[5], S. Pogodin[12], V. Prasad[9,10], F. Z. Qi[1], H. Qi[9,10], H. R. Qi[26], M. Qi[25], T. Y. Qi[44,89], S. Qian[1,9], W. B. Qian[21], Z. Qian[36], C. F. Qiao[21], J. J. Qin[67], L. Q. Qin[63], X. P. Qin[44,89], X. S. Qin[29], Z. H. Qin[1,9], J. F. Qiu[1], S. Q. Qu[26], K. H. Rashid[49], C. F. Redmer[5], K. J. Ren[8], A. Rivetti[7], V. Rodin[54], M. Rolo[7], G. Rong[1,21], Ch. Rosner[31], S. N. Ruan[23], A. Sarantsev[12,95], Y. Schelhaas[5], C. Schnier[4], K. Schoenning[3], M. Scodeggio[14,38], K. Y. Shan[44,89], W. Shan[74], X. Y. Shan[9,10], J. F. Shangguan[24], L. G. Shao[1,21], M. Shao[9,10], C. P. Shen[44,89], H. F. Shen[1,21], W. H. Shen[21], X. Y. Shen[1,21], B. A. Shi[21], H. C. Shi[9,10], J. Y. Shi[1], Q. Q. Shi[24], R. S. Shi[1,21], X. Shi[1,9], J. J. Song[45], W. M. Song[1,32], Y. X. Song[15,86], S. Sosio[6,7], S. Spataro[6,7], F. Stieler[5], P. P. Su[24], Y. J. Su[21], G. X. Sun[1], H. Sun[21], H. K. Sun[1], J. F. Sun[45], L. Sun[20], S. S. Sun[1,21], T. Sun[1,21], W. Y. Sun[32], Y. J. Sun[9,10], Y. Z. Sun[1], Z. T. Sun[29], Y. X. Tan[9,10], C. J. Tang[62], G. Y. Tang[1], J. Tang[36], L. Y. Tao[67], Q. T. Tao[28,88], M. Tat[37], J. X. Teng[9,10], V. Thoren[3], W. H. Tian[59], Y. Tian[21,27], I. Uman[75], B. Wang[1], B. Wang[9,10], B. L. Wang[21], C. W. Wang[25], D. Y. Wang[15,86], F. Wang[67], H. J. Wang[61,91,92], H. P. Wang[1,21], K. Wang[1,9], L. L. Wang[1], M. Wang[29], M. Z. Wang[15,86], Meng Wang[1,21], S. Wang[63], S. Wang[44,89], T. Wang[44,89], T. J. Wang[23], W. Wang[36], W. H. Wang[20], W. P. Wang[9,10], X. Wang[15,86], X. F. Wang[61,91,92], X. L. Wang[44,89], Y. Wang[26], Y. D. Wang[76], Y. F. Wang[1,9,21], Y. H. Wang[58], Y. Q. Wang[1], Yaqian Wang[1,77], Z. Wang[1,9], Z. Y. Wang[1,21], Ziyi Wang[21], D. H. Wei[63], F. Weidner[18], S. P. Wen[1], D. J. White[39], U. Wiedner[4], G. Wilkinson[37], M. Wolke[3], L. Wollenberg[4], J. F. Wu[1,21], L. H. Wu[1], L. J. Wu[1,21], X. Wu[44,89], X. H. Wu[32], Y. Wu[10], Y. J. Wu[27], Z. Wu[1,9], L. Xia[9,10], T. Xiang[15,86], D. Xiao[61,91,92], G. Y. Xiao[25], H. Xiao[44,89], S. Y. Xiao[1], Y. L. Xiao[44,89], Z. J. Xiao[42], C. Xie[25], X. H. Xie[15,86], Y. Xie[29], Y. G. Xie[1,9], Y. H. Xie[65], Z. P. Xie[9,10], T. Y. Xing[1,21], C. F. Xu[1,21], C. J. Xu[36], G. F. Xu[1], H. Y. Xu[52], Q. J. Xu[78], X. P. Xu[24], Y. C. Xu[79], Z. P. Xu[25], F. Yan[44,89], L. Yan[44,89], W. B. Yan[9,10], W. C. Yan[34], H. J. Yang[80,96], H. L. Yang[32], H. X. Yang[1], S. L. Yang[1,21], Tao Yang[1], Y. F. Yang[23], Y. X. Yang[1,21], Yifan Yang[1,21], M. Ye[1,9], M. H. Ye[81], J. H. Yin[1], Z. Y. You[36], B. X. Yu[1,9,21], C. X. Yu[23], G. Yu[1,21], T. Yu[67], X. D. Yu[15,86], C. Z. Yuan[1,21], L. Yuan[82], S. C. Yuan[1], X. Q. Yuan[1], Y. Yuan[1,21], Z. Y. Yuan[36], C. X. Yue[8], A. A. Zafar[49], F. R. Zeng[29], X. Zeng[65], Y. Zeng[28,88], X. Y. Zhai[32], Y. H. Zhan[36], A. Q. Zhang[1,21], B. L. Zhang[1,21], B. X. Zhang[1], D. H. Zhang[23], G. Y. Zhang[45], H. Zhang[10], H. H. Zhang[32], H. H. Zhang[36], H. Q. Zhang[1,9,21], H. Y. Zhang[1,9], J. L. Zhang[83], J. Q. Zhang[42], J. W. Zhang[1,9,21], J. X. Zhang[61,91,92], J. Y. Zhang[1], J. Z. Zhang[1,21], Jianyu Zhang[1,21], Jiawei Zhang[1,21], L. M. Zhang[26], L. Q. Zhang[36], Lei Zhang[25], P. Zhang[1], Q. Y. Zhang[8,34], Shuihan Zhang[1,21], Shulei Zhang[28,88], X. D. Zhang[76], X. M. Zhang[1], X. Y. Zhang[29], X. Y. Zhang[24], Y. Zhang[37], Y. T. Zhang[34], Y. H. Zhang[1,9], Yan Zhang[9,10], Yao Zhang[1], Z. H. Zhang[1], Z. L. Zhang[32], Z. Y. Zhang[23], Z. Y. Zhang[20], G. Zhao[1], J. Zhao[8], J. Y. Zhao[1,21], J. Z. Zhao[1,9], Lei Zhao[9,10], Ling Zhao[1], M. G. Zhao[23], S. J. Zhao[34], Y. B. Zhao[1,9], Y. X. Zhao[21,27], Z. G. Zhao[9,10], A. Zhemchugov[12,87], B. Zheng[67], J. P. Zheng[1,9], W. J. Zheng ⓘ[1,21], Y. H. Zheng[21], B. Zhong[42], C. Zhong[67], X. Zhong[36], H. Zhou[29], L. P. Zhou[1,21], X. Zhou[20], X. K. Zhou[21], X. R. Zhou[9,10], X. Y. Zhou[8], Y. Z. Zhou[44,89], J. Zhu[23], K. Zhu[1], K. J. Zhu[1,9,21], L. X. Zhu[21], S. H. Zhu[84], S. Q. Zhu[25], T. J. Zhu[83], W. J. Zhu[44,89], Y. C. Zhu[9,10], Z. A. Zhu[1,21], J. H. Zou[1] & J. Zu[9,10]

[1]Institute of High Energy Physics, Beijing 100049, People's Republic of China. [2]G.I. Budker Institute of Nuclear Physics SB RAS (BINP), Novosibirsk 630090, Russia. [3]Uppsala University, Box 516, SE-75120 Uppsala, Sweden. [4]Bochum Ruhr-University, D-44780 Bochum, Germany. [5]Johannes Gutenberg University of Mainz, Johann-Joachim-Becher-Weg 45, D-55099 Mainz, Germany. [6]University of Turin and INFN, University of Turin, I-10125 Turin, Italy. [7]INFN, I-10125 Turin, Italy. [8]Liaoning Normal University, Dalian 116029, People's Republic of China. [9]State Key Laboratory of Particle Detection and Electronics, Beijing 100049, Hefei 230026, People's Republic of China. [10]University of Science and Technology of China, Hefei 230026, People's Republic of China. [11]Southeast University, Nanjing 211100, People's Republic of China. [12]Joint Institute for Nuclear Research, 141980 Dubna, Moscow region, Russia. [13]INFN Laboratori Nazionali di Frascati, INFN Laboratori Nazionali di Frascati, I-00044 Frascati, Italy. [14]INFN Sezione di Ferrara, INFN Sezione di Ferrara, I-44122 Ferrara, Italy. [15]Peking University, Beijing 100871, People's Republic of China. [16]National Centre for Nuclear Research, Warsaw 02-093, Poland. [17]Institute of Physics and Technology, Peace Avenue 54B, Ulaanbaatar 13330, Mongolia. [18]University of Muenster, Wilhelm-Klemm-Strasse 9, 48149 Muenster, Germany. [19]Carnegie Mellon University, Pittsburgh, Pennsylvania 15213, USA. [20]Wuhan University, Wuhan 430072, People's Republic of China. [21]University of Chinese Academy of Sciences, Beijing 100049, People's Republic of China. [22]Turkish Accelerator Center Particle Factory Group, Istinye University, 34010 Istanbul, Turkey. [23]Nankai University, Tianjin 300071, People's Republic of China. [24]Soochow University, Suzhou 215006, People's Republic of China. [25]Nanjing University, Nanjing 210093, People's Republic of China. [26]Tsinghua University, Beijing 100084, People's Republic of China. [27]Institute of Modern Physics, Lanzhou 730000, People's Republic of China. [28]Hunan University, Changsha 410082, People's Republic of China. [29]Shandong University, Jinan 250100, People's Republic of China. [30]Yunnan University, Kunming 650500, People's Republic of China. [31]Helmholtz Institute Mainz, Staudinger Weg 18, D-55099 Mainz, Germany. [32]Jilin University, Changchun 130012, People's Republic of China. [33]Liaoning University, Shenyang 110036, People's Republic of China. [34]Zhengzhou University, Zhengzhou 450001, People's Republic of China. [35]University of Eastern Piedmont, I-15121 Alessandria, Italy. [36]Sun Yat-Sen University, Guangzhou 510275, People's Republic of China. [37]University of Oxford, Keble Road, Oxford OX13RH, UK. [38]University of Ferrara, I-44122 Ferrara, Italy. [39]University of Manchester, Oxford Road, Manchester M13 9PL, UK. [40]GSI Helmholtzcentre for Heavy Ion Research GmbH, D-64291 Darmstadt, Germany. [41]Guangxi University, Nanning 530004, People's Republic of China. [42]Nanjing Normal University, Nanjing 210023, People's Republic of China. [43]Shandong Normal University, Jinan 250014, People's Republic of China. [44]Fudan University, Shanghai 200433, People's Republic of China. [45]Henan Normal University, Xinxiang 453007, People's Republic of China. [46]University of Hawaii, Honolulu, Hawaii 96822, USA. [47]Suranaree University of Technology, University Avenue 111, Nakhon Ratchasima 30000, Thailand. [48]South China Normal University, Guangzhou 510006, People's Republic of China. [49]University of the Punjab, Lahore 54590, Pakistan. [50]Indiana University, Bloomington, Indiana 47405, USA. [51]Huangshan College, Huangshan 245000, People's Republic of China. [52]University of Jinan, Jinan 250022, People's Republic of China. [53]Instituto de Alta Investigación, Universidad de Tarapacá, Casilla 7D, Arica, Chile. [54]University of Groningen, NL-9747 AA Groningen, The Netherlands. [55]China University of Geosciences, Wuhan 430074, People's Republic of China. [56]Justus-Liebig-Universitaet Giessen, II. Physikalisches Institut, Heinrich-Buff-Ring 16, D-35392 Giessen, Germany. [57]Indian Institute of Technology Madras, Chennai 600036, India. [58]Qufu Normal University, Qufu 273165, People's Republic of China. [59]Shanxi Normal University, Linfen 041004, People's Republic of China. [60]Beijing Institute of Petrochemical Technology, Beijing 102617, People's Republic of China. [61]Lanzhou University, Lanzhou 730000, People's Republic of China. [62]Sichuan University, Chengdu 610064, People's Republic of China. [63]Guangxi Normal University, Guilin 541004, People's Republic of China. [64]Shanxi University, Taiyuan 030006, People's Republic of China. [65]Central China Normal University, Wuhan 430079, People's Republic of China. [66]Henan University of Science and Technology, Luoyang 471003, People's Republic of China. [67]University of South China, Hengyang 421001, People's Republic of China. [68]Henan University of Technology, Zhengzhou 450001, People's Republic of China. [69]Central South University, Changsha 410083, People's Republic of China. [70]Zhejiang University, Hangzhou 310027, People's Republic of China. [71]INFN Sezione di Perugia, I-06100 Perugia, Italy. [72]COMSATS University Islamabad, Lahore Campus, Defence Road, Off Raiwind Road, 54000 Lahore, Pakistan. [73]University of Perugia, I-06100 Perugia, Italy. [74]Hunan Normal University, Changsha 410081, People's Republic of China. [75]Near East University, Nicosia, North Cyprus, Mersin 10, Turkey. [76]North China Electric Power University, Beijing 102206, People's Republic of China. [77]Hebei University, Baoding 071002, People's Republic of China. [78]Hangzhou Normal University, Hangzhou 310036, People's Republic of China. [79]Yantai University, Yantai 264005, People's Republic of China. [80]Shanghai Jiao Tong University, Shanghai 200240, People's Republic of China. [81]China Center of Advanced Science and Technology, Beijing 100190, People's Republic of China. [82]Beihang University, Beijing 100191, People's Republic of China. [83]Xinyang Normal University, Xinyang 464000, People's Republic of China. [84]University of Science and Technology Liaoning, Anshan 114051, People's Republic of China. [85]the Novosibirsk State University, Novosibirsk 630090, Russia. [86]State Key Laboratory of Nuclear Physics and Technology, Peking University, Beijing 100871, People's Republic of China. [87]the Moscow Institute of Physics and Technology, Moscow 141700, Russia. [88]School of Physics and Electronics, Hunan University, Changsha 410082, China. [89]Key Laboratory of Nuclear Physics and Ion-beam Application (MOE) and Institute of Modern Physics, Fudan University, Shanghai 200443, People's Republic of China. [90]Guangdong Provincial Key Laboratory of Nuclear Science, Institute of Quantum Matter, South China Normal University, Guangzhou 510006, China. [91]Lanzhou Center for Theoretical Physics, Lanzhou University, Lanzhou 730000, People's Republic of China. [92]Frontiers Science Center for Rare Isotopes, Lanzhou University, Lanzhou 730000, People's Republic of China. [93]Goethe University Frankfurt, 60323 Frankfurt am Main, Germany. [94]the Department of Mathematical Sciences, IBA, Karachi, Pakistan. [95]the NRC "Kurchatov Institute", PNPI, 188300 Gatchina, Russia. [96]Key Laboratory for Particle Physics, Astrophysics and Cosmology, Ministry of Education; Shanghai Key Laboratory for Particle Physics and Cosmology; Institute of Nuclear and Particle Physics, Shanghai 200240, People's Republic of China. ✉e-mail: besiii-publications@ihep.ac.cn

