## [Peer Review File · Nature Communications]

Extracting the femtometer structure of strange baryons using the vacuum polarization effectREVIEWER COMMENTS

Reviewer #1 (Remarks to the Author):

The noteworthy results are measurements of a form factor ratio and phases for $\Lambda_{\bar{b}}$ Σ^0 baryon pairs. These results depend on the use of quantum correlations and the interference of an electromagnetic decay with a loop (or vacuum polarization contribution). The phase obtained is a relative phase between the electric and magnetic form factors, G_E and G_M , respectively.

The work is original and interesting but may not have as much impact as BESIII's previous Nature publications, which described the potential of baryon-antibaryon pairs produced at the J/ψ resonance for CP violation and related measurements. The paper is quite technically challenging and many of the normal details are placed in a section after the main body of the paper or in supplementary material.

For better understanding of the result, could the authors explicitly show or write down the correlated (or entangled) wavefunction of the baryon-antibaryon system ?

There may be a systematic issue in Figs 5d and 4d. Can the authors explicitly address this point ?

There are a number of writing issues (or editorial comments). These are listed below.

1 Title "using the vacuum polarization effect". I suppose that what is meant is the interference of Figs 1b) and 1c) rather than Fig 1b) by itself.

L5 "to gain a microscopic understanding" (missing word)

L12 "due to the effect of vacuum polarization..."

L15 "encoded in their form factor ratio and their phase"

L27 comma after "electromagnetic form factors"

L41 "carrying information about their space-like structure"

L68 explain the entanglement mentioned here

L96 "We also performed a measurement..."
L121 "Therefore, a simultaneous measurement...."
L122 "to explore""of exploring"
L127 "effect on fit results"
L150 "It""This"
L153 comma after "each other"
L156 "even greater sensitivity" or "even better sensitivity"
L257 "performed""used"
L268 "To suppress background noise" or "To suppress background"
L324 "of""for"
L325 "indicates transverse hyperon polarization"
L361-364 Shouldn't the results be posted to HEPData ?

Reviewer #2 (Remarks to the Author):

Title: Novel method to extract the femtometer structure of strange baryons using the vacuum polarization effect
Corresponding Author: Wenjing Zheng
Ms ID: NCOMMS-23-43432-T

The BESIII collaboration is pursuing a long-term program to measure the time-like form factors of the lowest lying baryons, with a recent emphasis of the instable hyperons. This is program very visible and of high quality.

In this work, the focus is on the Λ_{b1} - Σ (+ c.c.) final-state, triggered by some theoretical work in [23]. The main idea is discussed and shown in Fig.1, with a strong emphasis on the vacuum polarization contribution in Fig.1(b). In find this part of the work not well presented, for the following reasons:

- It is stated that the three-gluon vertex 1(a) is absent due to isospin

contribution of the strong interactions. However, in general isospin violation from the quark mass difference $m_u - m_d$ is of the same size as the one generated by the electromagnetic interaction. So there is no point in ignoring the strong isospin violation here. The authors must give a naive dimensional analysis (NDA) estimate of the three contributions and they will find that (b) dominates on the resonance.

- There strong emphasis on the vacuum polarization (VP) is, to my opinion, misleading. In fact, VP is given for all possible in/out momenta, what is relevant here is that one sits exactly on the J/ψ resonance, otherwise (b) would be suppressed by $\alpha/\pi \sim 1/400$. This needs to be stated more clearly.

- Fig.1 does not well represent the physics, the lines for the photon and the J/ψ need to be different.

Then, in the second part, use existing formalism to extract the relative phase and magnitude of the $\Lambda_{b0} - \Sigma_0$ form factor. This gives relevant new information, however, as this is the form factor ratio at one specific q^2 , it is not so clear what one really learns about the inner structure of the involved hyperons. Also, in this context the work by Buttimore and Jenkins in Eur. Phys. J. A 31 (2007) 9-14 should be quoted.

The third, and arguably most interesting part, is the test of CP symmetry by comparing $\Lambda_{b0} - \Sigma_0$ and its charged conjugated mode. This is novel. The result is consistent with zero and at present dominated by the statistical error. It would be good to know more precisely what sensitivity to direct CP violation BESIII will be able to deliver in the future.

Concerning the references, these are very selective. With respect to the proton form factors, it would be appropriate to additionally quote

Gao and Vanderhaeghen, Rev. Mod. Phys. 94 (2022) 1, 015002 as well as Lin, Hammer, Meissner, Phys. Rev. Lett. 128 (2022) 5, 052002.

In summary, this manuscript contains some interesting new information on the hyperon structure and direct CP violation in such baryon-antibaryon production processes. As stated, some more estimates on the various production processes are required to make the first part more amenable to a general readership. Only after that a decision on the suitability of this manuscript for Nature Communications can be made.

Reviewer #3 (Remarks to the Author):

The production of J/ψ hadrons exactly at the resonance energy allows for measurements where all decay products are analysed. The quantum correlated state of all particles that are formed, combined with the exact knowledge of the initial state, allows for a set of measurements that otherwise would be impossible to perform. This paper represents the first time that this quantum correlation is exploited with a pair of particles in the final state that are not each other's antiparticles. It is novel as the asymmetric final state allows to isolate only electromagnetic processes for the production and thus extract form factor information. In addition it also allows for a search of direct CP violation in the process which with the symmetric final state is not possible.

The work is original and will further the understanding of the structure of baryons which are the main building blocks of the Universe. The measurement of no CP violation is interesting but it should be explained that CP violation would require new physics as the Standard Model process investigated is purely electromagnetic where there is no CP violation.

The overall methodology is clear and the conclusions are solid. A very interesting paper. However, in several places, it is unclear exactly what the process is. While there are no prospects of replicating the exact measurement elsewhere at the moment, there should be clarity of the process.

l84: The integrated luminosity of the sideband data is discussed, but as the integrated luminosity used at the J/ψ resonance is never mentioned it is impossible to judge the statistical accuracy of this sideband measurement.

l109: The sentence starting "The probability" is completely unclear. Does it refer to "The probability density function"?

l119: The notation of Φ_1 and Φ_2 is confusing. It would be much better to label them with " $\bar{\Lambda}\Sigma$ " and " $\Lambda\bar{\Sigma}$ " as the subscripts. This would make the abstract easier to understand as well.

Fig 3 (right): There is a huge variation in the uncertainty of the C_{xz} parameter for different bins of \cos_θ . Given the "mirrored" effect of this for the red and blue curve it seems to be related to a detector effect. Can this be explained?

Are the references [8-18] that refer to proceedings really required? Usually in particle physics, the information in proceedings will be full covered by journal publications and there is no need to separately refer to proceedings.

Suppl. Fig 3: The label for the Green line is confusing. It should just be called "Background".

Suppl. Fig 4: It should be explained why the global fit line is jagged. This presumably come from that the background likelihood function in Eq. (7) is somehow binned in the angles, but this is not explained anywhere.

Suppl. Table 1: It is very confusing to quote the systematic uncertainty of a phase measurement as a relative uncertainty. It makes it appear like there is a different systematic uncertainty for $\Delta\Phi_1$ and $\Delta\Phi_2$, but in reality that is not the case. The systematic uncertainties for those should just be quoted in radians instead.

Best regards,

Ulrik Egede.

Reviewer 1

The noteworthy results are measurements of a form factor ratio and phases for $\bar{\Lambda}\Sigma^0$ baryon pairs. These results depend on the use of quantum correlations and the interference of an electromagnetic decay with a loop (or vacuum polarization contribution). The phase obtained is a relative phase between the electric and magnetic form factors, G_E and G_M , respectively.

The work is original and interesting but may not have as much impact as BESIII's previous Nature publications, which described the potential of baryon-antibaryon pairs produced at the J/ψ resonance for CP violation and related measurements. The paper is quite technically challenging and many of the normal details are placed in a section after the main body of the paper or in supplementary material.

For better understanding of the result, could the authors explicitly show or write down the correlated (or entangled) wavefunction of the baryon-antibaryon system?

A: $J/\psi \rightarrow B_1\bar{B}_2$ can be described by $\langle N^{h_1}(p_1, s_1)\bar{N}^{\bar{h}_2}(p_2, s_2)|J_\mu(0)|0\rangle = \bar{u}(p_1, s_1)\Gamma_\mu v(p_2, s_2)$, where $\Gamma_\mu = \left(F_1\gamma_\mu + F_2\frac{i\sigma_{\mu\nu}q^\nu}{2m}\right)$. In fact, the correlation between B_1 and \bar{B}_2 arises from the vertex form factors. As a result, the spins of B_1 and \bar{B}_2 are entangled, with the coefficient of this entanglement represented by $C_{\mu\nu}$ in the manuscript, which signifies the polarization correlation. Since these were covered in the reference [PhysRevD.16.2165, PhysRevD.99.056008], we did not provide the detailed information in the manuscript.

There may be a systematic issue in Figs 5d and 4d. Can the authors explicitly address this point?

A: In fact, "Figs 4 and 5" represent the fit results in moments, which are directly related to the amplitude. The most fundamental fit variables in the amplitude are the helicity angular distributions of the final-state particles, as shown in the Supplementary Figs.4f and 5f in the updated manuscript, where we can see that the fit results agree well with the data. We then transferred these angular variables to the moments using Eq. 8 and propagated the uncertainties accordingly.

There are a number of writing issues (or editorial comments). These are listed below.

1. Title "using the vacuum polarization effect". I suppose that what is meant is the interference of Figs 1b) and 1c) rather than Fig 1b) by itself.

A: Actually, Fig 1b) describes the process of $e^+e^- \rightarrow \bar{\Lambda}\Sigma^0$ through vacuum polarization effect at J/ψ . However, since Figs 1b) and 1c) have the same interaction vertex of $J/\psi \rightarrow \bar{\Lambda}\Sigma^0$, they have the same form factors. Therefore, when we extract decay parameters from Figs 1b), the electromagnetic form factor of $e^+e^- \rightarrow \bar{\Lambda}\Sigma^0$ is naturally obtained as well, which is same for Figs 1b) and 1c).

2. L5 "to gain a microscopic understanding" (missing word)

A: Thank you for the reminder. We have added it in the updated manuscript.

3. L12 "due to the effect of vacuum polarization..."

A: Many thanks for your suggestion! we updated this sentence in the updated manuscript.

4. L15 "encoded in their form factor ratio and their phase"

A: Many thanks for your suggestion! we updated this sentence in the updated manuscript.

5. L27 comma after "electromagnetic form factors"

A: Sorry for the negligence. The comma has been added after "electromagnetic form factors".

6. L41 "carrying information about their space-like structure"

A: Many thanks for your suggestion! we updated this sentence in the updated manuscript.

7. L68 explain the entanglement mentioned here

A: The majority of J/ψ baryonic decays proceeds via two-body intermediate states, where both baryon and anti-baryon are simultaneously produced. The spins of baryon and anti-baryon are not independent, but rather entangled. Since these were covered in the reference [Found Phys 11, 171–177 (1981), Sci Rep 5, 11591 (2015)], we did not provide the detailed information in the manuscript.

8. L96 "We also performed a measurement..."

A: Thank you for the reminder. We have revised it in the new version.

9. L121 "Therefore, a simultaneous measurement...."

A: Thank you for the reminder. We have revised it in the new version.

10. L122 "to explore" → "of exploring"

A: Many thanks! We updated the manuscript according to your suggestion.

11. L127 "effect on fit results"

A: Many thanks for point out this issue! We updated it accordingly in the updated manuscript.

12. L150 "It" → "This"

A: According to your suggestion, we updated the manuscript accordingly.

13. L153 comma after "each other"

A: Thanks for the reminder. The comma is inserted after "each other".

14. L156 "even greater sensitivity" or "even better sensitivity"

A: Many thanks! We updated the manuscript according to your suggestion.

15. L257 "performed" → "used"

A: Many thanks! We updated the manuscript according to your suggestion.

16. L268 "To suppress background noise" or "To suppress background"

A: According to your suggestion, we removed "noise" in the updated manuscript.

17. L324 "of" → "for"

A: Sorry for my improper expression. "For" is used instead of "of" in the new version.

18. L325 "indicates transverse hyperon polarization"

A: According to your suggestion, we updated the manuscript accordingly.

19. L361-364 Shouldn't the results be posted to HEPData?

A: At present, we did not submit the results to HEPData since maybe a slightly change based on the referees' suggestions/questions. After publication, we can post the results to HEPData. However, since this is a multi-dimensional fit, we are still discussing how to post the data points, including efficiency correction and uncertainty propagation, so that the readers can easily and directly use these data to fit the theoretical models by themselves. Of course, we would also like to help as they request.

Reviewer 2

The BESIII collaboration is pursuing a long-term program to measure the time-like form factors of the lowest lying baryons, with a recent emphasis of the instable hyperons. This is program very visible and of high quality.

In this work, the focus is on the $\bar{\Lambda}\Sigma + c.c.$ final-state, triggered by some theoretical work in [23]. The main idea is discussed and shown in Fig.1, with a strong emphasis on the vacuum polarization contribution in Fig.1(b). In find this part of the work not well presented, for the following reasons:

- It is stated that the three-gluon vertex 1(a) is absent due to isospin conservation of the strong interactions. However, in general isospin violation from the quark mass difference $m_u - m_d$ is of the same size as the one generated by the electromagnetic interaction. So there is no point in ignoring the strong isospin violation here. The authors must give a naive dimensional analysis (NDA) estimate of the three contributions and they will find that (b) dominates on the resonance.

A: Yes, the mass difference of $m_u - m_d$ also may lead to the isospin violation, but it not happens in the decay of $J/\psi \rightarrow \bar{\Lambda}\Sigma$. According to the theoretical discussions (e.g., arXiv: 1505.03930, arXiv: 1709. 09545, arXiv:2111.15045, arXiv: 2002.09675), this decay is only caused by the difference of the electric charge, which means that the mass difference of $m_u - m_d$ has no contribution in this decay. Therefore, the contribution from Fig. 1 (a) is believed to be absent in this decay. On the one hand, due to the fact that when a decay violates isospin the purely strong amplitude is suppressed by the small dimensionless factor ($\frac{m_u - m_d}{\sqrt{s}} \sim 10^{-3}$), as shown as Ref[arXiv: 2002.09675], and the fact that $\text{BF}(ggg)$ for J/ψ is 64% (PDG), the suppressed strong interaction is negligible with respect to γgg (8.8%) and γ^* (13%) mediated decays. Therefore, for isospin violating J/ψ decays, the process of ggg is suppressed. On the other hand, from Ref. [15](Eur. Phys. J. C 80, 903 (2020)), $\Lambda\bar{\Sigma}^0$ via γgg from J/ψ is also suppressed. Ref. [13] proposes the method of comparing the modulus values of the same electromagnetic coupling extracted from the branching ratio of $J/\psi \rightarrow \bar{\Lambda}\Sigma^0 + c.c.$ and the cross section of the reaction $e^+e^- \rightarrow \bar{\Lambda}\Sigma^0 + c.c.$ at the J/ψ mass. The fact that for the J/ψ the coupling is in good agreement with the value extracted from the cross section data, that depends only by the EM interaction, represents a clear indication that the process of γgg is negligible for the $J/\psi \rightarrow \Lambda\bar{\Sigma}^0$. However, γgg for $\psi' \rightarrow \Lambda\bar{\Sigma}^0$ is significant. For this interesting phenomenon, Ref. [15] has gives two different sources of isospin-violation. More details can be found in this reference.

- There strong emphasis on the vacuum polarization (VP) is, to my opinion, misleading. In fact, VP is given for all possible in/out momenta, what is relevant here is that one sits exactly on the J/ψ resonance, otherwise (b) would be suppressed by $\alpha/\pi \sim 1/400$. This needs to be stated more clearly.

A: Yes, you are right that the significant enhancement is indeed present due to the resonance at J/ψ . To clarify this issue, we revised the wording in the updated manuscript.

- Fig.1 does not well represent the physics, the lines for the photon and the J/ψ need to be different.

A: In Fig. 1, different line styles are indeed used for the photon and J/ψ . The photon is represented by a wave line, while the J/ψ is depicted by a fermion loop, signifying the hadronic vacuum polarization at the J/ψ .

Then, in the second part, use existing formalism to extract the relative phase and magnitude of the $\bar{\Lambda}\Sigma^0$ form factor. This gives relevant new information, however, as this is the form factor ratio at one specific q^2 , it is not so clear what one really learns about the inner structure of the involved hyperons. Also, in this context the work by Buttimore and Jenkins in Eur. Phys. J. A 31 (2007) 9-14 should be quoted.

A: The study of the internal structure of hyperons has always been one of the important research topics in

particle physics. The electromagnetic form factor is an important physical quantity that reflects the internal dynamics of hyperons. However, due to the instability of hyperons, their electromagnetic form factors cannot be extracted using traditional electron scattering techniques as in the case of nucleons. Instead, their time-like form factors ($|G_E/G_M|$ and $\arg(G_E/G_M)$) can be obtained from electron-positron annihilation processes, which subsequently produce hyperon-antihyperon pairs. Ref. [25] (Phys. Rep. 550, 1 (2015)) illustrates the varying energy dependence of the ratios and phases of the electromagnetic form factors of different final-state hyperons. It seems that $|G_E/G_M|$ and $\arg(G_E/G_M)$ are very sensitive to the details of the final-state hyperons interaction. Therefore, it would be very interesting to perform further experiments that establish reliably the energy dependence of $|G_E/G_M|$ and $\arg(G_E/G_M)$. On the other hand, we could write that dispersive calculations (quote Phys. Rev. D 104, 116016) with these results in combination of those from our $e^+e^- \rightarrow \Lambda\bar{\Sigma}^0$ cross section paper (Phys. Rev. D 109, 012002), put constraints the structure functions in the space-like regions and hence they provide information on the inner structure. This is even more true of future experiments can provide measurements on the ratio R in more points. The reference you recommended has already been cited as Ref. [5] in the new manuscript.

The third, and arguably most interesting part, is the test of CP symmetry by comparing $\bar{\Lambda}\Sigma^0$ and its charged conjugated mode. This is novel. The result is consistent with zero and at present dominated by the statistical error. It would be good to know more precisely what sensitivity to direct CP violation BESIII will be able to deliver in the future.

A: Since the luminosity is expected to improve about 3 times after BEPCII/BESIII upgrade, the improvement on sensitivity at BESIII is not significant. However, a project of Super tau-charm facility (STCF) [Front. Phys. 19(1), 14701 (2024), arXiv: 2303.15790] is proposed in China, The designed luminosity is about 100 times larger than that of BEPCII, which enables one to collect unprecedented high statistics data samples. Therefore, in the future STCF opens the possibility to investigate the CP violation in this decay with an excellent sensitivity.

Concerning the references, these are very selective. With respect to the proton form factors, it would be appropriate to additionally quote Gao and Vanderhaeghen, Rev. Mod. Phys. 94 (2022) 1, 015002 as well as Lin, Hammer, Meissner, Phys. Rev. Lett. 128 (2022) 5, 052002.

A: Many thanks for suggestions. We added these two references as Ref. [6] and Ref. [7] in the updated manuscript.

In summary, this manuscript contains some interesting new information on the hyperon structure and direct CP violation in such baryon-antibaryon production processes. As stated, some more estimates on the various production processes are required to make the first part more amenable to a general readership. Only after that a decision on the suitability of this manuscript for Nature Communications can be made.

Reviewer 3

The production of J/ψ hadrons exactly at the resonance energy allows for measurements where all decay products are analysed. The quantum correlated state of all particle that are formed, combined with the exact knowledge of the initial state, allows for a set of measurements that otherwise would be impossible to perform. This paper represents the first time that this quantum correlation is exploited with a pair of particle in the final state that are not each others antiparticles. It is novel as the asymmetric final state allows to isolate only electromagnetic processes for the production and thus extract form factor information. In addition it also allows for a search of direct CP violation in the process which with the symmetric final state is not possible.

The work is original and will further the understanding of the structure of baryons which are the main building blocks of the Universe. The measurement of no CP violation is interesting but it should be explained that CP violation would require new physics as the Standard Model process investigated is purely electromagnetic where there is no CP violation.

A: Within the framework of the Standard Model, there is no CP violation in purely electromagnetic processes, which means that in this case, $\Delta\Phi_{CP}$ should be zero as mentioned in the manuscript. Since the manuscript mainly focus on the measurement of polarizations and the ratio of G_E/G_M , we did not talk too much about the search for CP violation. According to your suggestion, we rewrote the corresponding sentences in the updated manuscript to make clear for readers that these process are of interest for searching for the additional source of CP violation beyond the Standard Model.

The overall methodology is clear and the conclusions are solid. A very interesting paper. However, in several places, it is unclear exactly what the process is. While there are no prospects of replicating the exact measurement elsewhere at the moment, there should be clarity of the process.

L84: The integrated luminosity of the sideband data is discussed, but as the integrated luminosity used at the J/ψ resonance is never mentioned it is impossible to judge the statistical accuracy of this sideband measurement.

A: To clarify this issue, the integrated luminosity used at the J/ψ resonance, 3083 pb^{-1} from Ref. [17], is added in the updated manuscript.

L109: The sentence starting "The probability" is completely unclear. Does it refer to "The probability density function"?

A: Sorry for confusion, here "The probability" refers to "The feasibility of extracting the form factors", which was modified accordingly in the updated manuscript.

L119: The notation of Φ_1 and Φ_2 is confusing. It would be much better to label them with " $\bar{\Lambda}\Sigma$ " and " $\Lambda\bar{\Sigma}$ " as the subscripts. This would make the abstract easier to understand as well.

A: According to your suggestion, we labeled them with " $\bar{\Lambda}\Sigma$ " and " $\Lambda\bar{\Sigma}$ " and modified accordingly in the updated manuscript.

Fig 3 (right): There is a huge variation in the uncertainty of the C_{xz} parameter for different bins of $\cos\theta$. Given the "mirrored" effect of this for the red and blue curve it seems to be related to a detector effect. Can this be explained?

A: Actually, the error is obtained through error propagation, which has a direct relationship with $\sin\Delta\Phi$ and the error of $\Delta\Phi$. However, the fluctuations in different intervals and process are different. For example,

in the sixth blue point, the fluctuation in this bin results in a more smaller $\Delta\Phi$ with a larger error. This makes the error propagated to C_{xz} very small.

Are the references [8-18] that refer to proceedings really required? Usually in particle physics, the information in proceedings will be full covered by journal publications and there is no need to separately refer to proceedings.

A: According to your suggestions, we updated the references accordingly and removed the proceedings which already covered by the journal publications.

Suppl. Fig 3: The label for the Green line is confusing. It should just be called "Background".

A: Many thanks! To make it clear for readers, we updated the figure in the manuscript.

Suppl. Fig 4: It should be explained why the global fit line is jagged. This presumably come from that the background likelihood function in Eq. (7) is somehow binned in the angles, but this is not explained anywhere.

A: In fact, this fit does not simply involve fitting to the corresponding physical quantity distribution with a probability density function (in which case the fit result would be a smooth curve, as shown in Supplementary Fig. 2 and 3). Instead, we directly fit the four-momentum of the final-state particles. During this process, we weight the phase-space Monte Carlo sample events to match the data. After that, we can obtain weights for each event from fit. These weights allow us to derive the distributions of the weighted phase-space samples, which are the fit results we observe. Therefore, in this scenario, the global fit line appears jagged. To avoid confusing readers, we made an explanation in the updated manuscript.

Suppl. Table 1: It is very confusing to quote the systematic uncertainty of a phase measurement as a relative uncertainty. It makes it appear like there is a different systematic uncertainty for $\Delta\Phi_1$ and $\Delta\Phi_2$, but in reality that is not the case. The systematic uncertainties for those should just be quoted in radians instead.

A: Sorry for confusion. In fact, the relative uncertainty here refers to the proportion of the uncertainty relative to the central value. However, to avoid confusion for the readers, we have updated to absolute uncertainty, in units of radians.

REVIEWER COMMENTS

Reviewer #1 (Remarks to the Author):

This is a short second review.

The authors response on the role of quantum entanglement is not quite satisfactory. Instead of clearly explaining the role of QM entanglement in this result, the authors refer to some other PRD papers. There is similar issue with the discussion of the vacuum polarization contribution and the response to referee #2.

Why publish in Nature if these two core issues cannot be explained to the general physics reader ? If this is not possible, consider JHEP or Physical Review D where technical issues can be described in detail.

Reviewer #2 (Remarks to the Author):

Title: Novel method to extract the femtometer structure of strange baryons using the vacuum polarization effect, 1st revised version

Corresponding Author: Wenjing Zheng

Ms ID: NCOMMS-23-43432-T

I am still struggling with this manuscript. While the authors have taken into account some of my suggestions, there are still issues that need to be clarified:

- The authors make a big point about vacuum polarization. This is not the proper wording, the conversion of photons into vector mesons and the consequent coupling to other particles has been called photon-hadron interactions since long. See the well-known textbook by Richard Feynman, "Photon-hadron Interactions", CRC Press,

DOI <https://doi.org/10.1201/9780429493331>

So this needs to be changed throughout. Also, since they essentially use the well-known resonance enhancement here, is this really something new?

- The authors have only partially answered my query on the strong isospin violation effects. As demanded, for a general audience as in this journal, one has to provide an NDA argument why the strong isospin violation is so much suppressed compared to the electromagnetic ones. A mere reference to a number of papers is not sufficient because this is such a central issue.

- Similarly, there is no convincing answer why the form factor ratio at this particular momentum transfer is important. I don't need a lecture on the relevance of measuring hyperon form factors but rather concrete answers to my query. This must also be reflected in the manuscript.

In view of these comments, I can not recommend publication.

Reviewer #3 (Remarks to the Author):

The authors have taken my comments from the first review carefully into account. A minor comment about the 6th blue bin of Fig. 3 as the authors discuss in their reply. I wonder if there is a breakdown of the inherent linear assumption in error propagation for this bin. As the individually binned measurements here only act as a cross check this is not an issue for the overall result but the authors might nevertheless with toy MC studies or similar want to study if this uncertainty can be trusted.

I summary, fully support a publication in the current form of the manuscript.

Kind regards,
Ulrik Egede

Reviewer#1

The authors response on the role of quantum entanglement is not quite satisfactory. Instead of clearly explaining the role of QM entanglement in this result, the authors refer to some other PRD papers. There is similar issue with the discussion of the vacuum polarization contribution and the response to referee#2.

Re: In this work, "quantum entanglement" refers to the fact that two particles originating from a decay of another particle, are entangled which manifest in correlated spin. This use of "entanglement" in the context of a baryon and an antibaryon is fairly common in high energy physics (see e.g. Nature Physics volume 15, pages 631–634 (2019) [1]), though it is not strictly the same as how the word is used in quantum mechanics.

For the entanglement for the Λ and $\bar{\Lambda}$ in the final states, theoretically the joint-decay distributions of them can not directly written as a product of Λ and $\bar{\Lambda}$ distribution functions. That means that both of them entangled together in the decaying angular distributions. We may take the simplest case of $J/\psi \rightarrow \Lambda\bar{\Lambda}$ for example to illustrate this issue. The joint angular distribution of the decay chain $J/\psi \rightarrow (\Lambda \rightarrow p\pi^-)(\bar{\Lambda} \rightarrow \bar{p}\pi^+)$ can be expressed as

$$\begin{aligned} \mathcal{W}(\boldsymbol{\xi}; \alpha_\psi, \Delta\Phi, \alpha_-, \alpha_+) = & 1 + \alpha_\psi \cos^2 \theta_\Lambda \quad (\text{Unpolarized part}) \\ & + \alpha_- \alpha_+ [\sin^2 \theta_\Lambda (n_{1,x} n_{2,x} - \alpha_\psi n_{1,y} n_{2,y}) + (\cos^2 \theta_\Lambda + \alpha_\psi) n_{1,z} n_{2,z}] \\ & + \alpha_- \alpha_+ \sqrt{1 - \alpha_\psi^2} \cos(\Delta\Phi) \sin \theta_\Lambda \cos \theta_\Lambda (n_{1,x} n_{2,z} + n_{1,z} n_{2,x}) \quad (\text{Entangled, or spin correlated, part}) \\ & + \sqrt{1 - \alpha_\psi^2} \sin(\Delta\Phi) \sin \theta_\Lambda \cos \theta_\Lambda (\alpha_- n_{1,y} + \alpha_+ n_{2,y}) \quad (\text{Polarized part}), \end{aligned} \quad (1)$$

where $\hat{\mathbf{n}}_1$ ($\hat{\mathbf{n}}_2$) is the unit vector in the direction of the nucleon (antinucleon) in the rest frame of Λ ($\bar{\Lambda}$). The components of these vectors are expressed using a common $(\hat{x}, \hat{y}, \hat{z})$ coordinate system with the orientation shown in Fig. 1. The \hat{z} axis in the Λ and $\bar{\Lambda}$ rest frames is oriented along the Λ momentum \mathbf{p}_Λ in the J/ψ rest system. The \hat{y} axis is perpendicular to the reaction plane and oriented along the vector $\mathbf{k}_- \times \mathbf{p}_\Lambda$, where \mathbf{k}_- is the electron beam momentum in the J/ψ rest system. The variable $\boldsymbol{\xi}$ denotes the tuple $(\theta_\Lambda, \hat{\mathbf{n}}_1, \hat{\mathbf{n}}_2)$, a set of kinematic variables which uniquely specify an event configuration.

FIG. 1. Kinematics of the reaction $e^+e^- \rightarrow J/\psi \rightarrow \Lambda\bar{\Lambda}$ in the overall center-of-mass system. The Λ particle is emitted in the \hat{z} direction at an angle θ_Λ with respect to the e^- direction, and the $\bar{\Lambda}$ is emitted in the opposite direction. The hyperons are polarized in the direction perpendicular to the reaction plane (\hat{y}). The hyperons are reconstructed, and the polarization is determined by measuring their decay products: (anti-)nucleons and pions.

In the above Eq. (1), the terms multiplied by $\alpha_- \alpha_+$ represent the contribution from $\Lambda\bar{\Lambda}$ spin correlations, which means that Λ and $\bar{\Lambda}$ are entangled. To avoid confusing the concept of "quantum entanglement" in Quantum computing or Quantum information, we would like to replace "entanglement" with "spin correlation". While the terms multiplied by α_- and α_+ separately represent the contribution from the polarization. The presence of all three contributions in Eq. (1) enables an unambiguous determination of the parameters α_ψ and $\Delta\Phi$ and the decay asymmetries α_- , α_+ . This is the advantage from the quantum spin correlation

between baryon and antibaryon [1].

Regarding the contribution of vacuum polarization, in the process of $e^+e^- \rightarrow \gamma^* \rightarrow c\bar{c}$ (*loop*) $\rightarrow \gamma^* \rightarrow B\bar{B}$, the vacuum polarization represents the resonance enhancement effect at the J/ψ . This leads to a large cross-section at 3.097 GeV, which greatly increases the number of signal events we can observe, resulting in smaller uncertainties. Moreover, the vacuum polarization implies that J/ψ ($c\bar{c}$) $\rightarrow B\bar{B}$ is a pure electromagnetic process with single-photon exchange, which has the same $\gamma^* \bar{\Lambda}\Sigma^0$ vertex as $e^+e^- \rightarrow \gamma^* \rightarrow B\bar{B}$. Therefore, the form factor at the $\gamma^* \bar{\Lambda}\Sigma^0$ vertex can be directly extract from $e^+e^- \rightarrow \gamma^* \rightarrow c\bar{c}$ (*loop*) $\rightarrow \gamma^* \rightarrow B\bar{B}$ by measuring the polarization of final baryon-antibaryon pair with high statistics and low uncertainties. Also, "vacuum polarization" is a frequently used concept and wording in hadron physics, in particularly in the context of form factors and hadronic corrections to the muon $g - 2$. For this issue, we shall also try to make a clear response to Referee #2.

REFERENCES

- [1] Ablikim, M. et al. Polarization and entanglement in baryon–antibaryon pair production in electron–positron annihilation. *Nature Phys.* **15**, 631 (2019).

Reviewer#2

I am still struggling with this manuscript. While the authors have taken into account some of my suggestions, there are still issues that need to be clarified:

- The authors make a big point about vacuum polarization. This is not the proper wording, the conversion of photons into vector mesons and the consequent coupling to other particles has been called photon-hadron interactions since long. See the well-known textbook by Richard Feynman, "Photon-hadron Interactions", CRC Press, DOI <https://doi.org/10.1201/9780429493331>. So this needs to be changed throughout. Also, since they essentially use the well-known resonance enhancement here, is this really something new?

Re: In this work, it is important to distinguish the general hadron-photon interactions (which include all diagrams Fig. 1(a)-1(d)). Vacuum polarization only denotes diagram Fig. 1(c), i.e. when the virtual photon fluctuates into an intermediate state and then back to the photon state. In hadron physics, especially in the context of meson form factors and hadronic corrections to the muon $g - 2$, vacuum polarization very frequently used to this day. What is novel about this work is to exploit vacuum polarization for precise measurements of hyperon form factors. This has not been done before, to our best knowledge.

For example, for the J/ψ decaying into other hadronic final states, not only just single-photon exchange, but the gluons are involved. This means that the $\gamma^* \rightarrow c\bar{c}$ (loop) $\rightarrow ggg$ and $\gamma \rightarrow c\bar{c}$ (loop) $\rightarrow \gamma^* gg$ process occur, as shown in Fig. 1(a) and 1(b), which is not the vacuum polarization, but simply the interaction between photons and hadrons and the resonance enhancement.

For the decay of interested $J/\psi \rightarrow \bar{\Lambda}\Sigma^0$ in the manuscript, it is a pure electromagnetic process proceeding with single-photon exchange [1, 2] and the J/ψ particle is specifically composed of $c\bar{c}$, proceeding with the $\gamma^* \rightarrow c\bar{c}$ (loop) $\rightarrow \gamma^*$ process as shown in Fig. 1(c), which is the so called the hadronic vacuum polarization effect. This effect succinctly and accurately helps us pinpoint the potential mechanism of this process and distinguished from the above two cases illustrated in Figs. 1(a) and 1(b). Comparing to the contribution directly from electron-positron annihilation displayed in Fig. 1(d), it is precisely because of this hadronic vacuum polarization process makes the signal statistics is significantly increased, and the corresponding statistical uncertainty is reduced.

More importantly, since the timelike form factors need to be extracted from single-photon exchange process, i.e. $e^+e^- \rightarrow \gamma^* \rightarrow \bar{\Lambda}\Sigma^0$ as shown in Fig. 1(d), and $e^+e^- \rightarrow \gamma^* \rightarrow c\bar{c}$ (loop) $\rightarrow \gamma^* \rightarrow \bar{\Lambda}\Sigma^0$ and $e^+e^- \rightarrow \gamma^* \rightarrow \bar{\Lambda}\Sigma^0$ have the same $\gamma^* \bar{\Lambda}\Sigma^0$ vertex, this allows us to directly extract the form factor at the $\gamma^* \bar{\Lambda}\Sigma^0$ vertex from $e^+e^- \rightarrow \gamma^* \rightarrow c\bar{c}$ (loop) $\rightarrow \gamma^* \rightarrow \bar{\Lambda}\Sigma^0$, with the advantage of high statistics.

FIG. 1. **The Feynman diagrams for $e^+e^- \rightarrow \text{hadrons}$ in the vicinity of the J/ψ .** (a) strong process with intermediate J/ψ mediated by gluons (ggg), (b) the mixed strong-electromagnetic process of J/ψ decay mediated by ggg , (c) electromagnetic process through the vacuum polarization of one virtual photon (γ^*) to J/ψ , (d) continuum process without the J/ψ intermediate state but only one virtual photon.

- The authors have only partially answered my query on the strong isospin violation effects. As demanded, for a general audience as in this journal, one has to provide an NDA argument why the strong isospin violation is so much suppressed compared to the electromagnetic ones. A mere reference to a number of papers is not sufficient because this is such a central issue.

Re: According to your suggestion, we may try to clarify this issue with the naive dimensional analysis as

described below:

After discussions with theorists, we are suggested to provide the simplest example, the mass difference between proton and neutron, for the isospin violation caused by the mass difference of u and d quarks. As we know, the proton, composed of uud quarks, has a mass of 938.27 MeV, while the neutron, composed of udd quarks, has a mass of 939.57 MeV. The mass difference between the u and d quarks can be approximately determined by the mass difference between the proton and neutron, which is only about 0.001 GeV. Therefore, the isospin breaking in the strong interaction caused by the mass difference of the u and d quarks is very small, although we are currently unable to quantitatively calculate this specific suppression due to its highly complex and unknown intermediate processes.

For the J/ψ decays, the dominant contributions are from strong interaction (ggg), pure electromagnetic interaction (γ^*), and the interference of strong and electromagnetic interactions (γgg) with approximately branching fractions of 64.1%, 13.5%, and 8.8% respectively, as displayed in Fig. 1, in accordance with the Particle Data Group (PDG) [3]. However, in case of the isospin violation in J/ψ decays, the contribution from the strong interaction (i.e., the ggg process) will be suppressed by the small dimensionless factor $\frac{m_d - m_u}{\sqrt{s}} \sim 10^{-3}$ (here m_u and m_d are the masses of u and d quarks, \sqrt{s} is the mass of J/ψ) [1]. After taking into account the suppression the factor $\frac{m_u - m_d}{\sqrt{s}} \sim 10^{-3}$, the branching fraction of the isospin violating decays of J/ψ through strong interaction (ggg) is approximately 0.0641%, which is greatly suppressed compared to the 13.5% branching fraction of electromagnetic interaction (γ^*). In this case, we take $J/\psi \rightarrow \bar{\Lambda}\Sigma^0$ as a pure electro-magnetic decay and ignored the contribution from the strong interaction in the manuscript. We have tried to clarify this in the new version of the paper, but that we are open for suggestions on how to make this even clearer to the reader.

- Similarly, there is no convincing answer why the form factor ratio at this particular momentum transfer is important. I don't need a lecture on the relevance of measuring hyperon form factors but rather concrete answers to my query. This must also be reflected in the manuscript.

Re: Due to the complexity of form factors in the time-like region, baryons, B , and anti-baryons, \bar{B} produced through the annihilation process $e^+e^- \rightarrow B\bar{B}$ are naturally polarized along the direction orthogonal to the scattering plane. This property is relevant in the case of the Λ baryon because the self-analyzing weak decay $\Lambda \rightarrow p\pi^-$ can be exploited to measure the Λ polarization vector, which is obtained from the angular distribution of the final proton and pion.

All in all, by measuring the differential cross section of an annihilation process $e^+e^- \rightarrow B\bar{B}$ and the polarization of final baryons and anti-baryons, we can obtain the moduli and the relative phase of electric and magnetic Sachs form factors of the baryon B . Since, form factors are analytic functions of q^2 , where q is the four-momentum of the virtual photon which mediates the annihilation in Born approximation and are defined in the whole q^2 -complex plane with a cut along the real axis from the so-called theoretical threshold $q^2 = (2m_\pi)^2$ (m_π is the charged pion mass) up to infinity, they can be investigated by using dispersive techniques, i.e., integral relations between their values in different kinematical domains, namely the space-like, $q^2 < 0$, and time-like, $q^2 > 0$, region.

Their ratio, rather than single form factors, is particularly suitable to be studied with dispersive approaches. Indeed, the ratio is completely known, we know the modulus and the phase, which corresponds to the relative phase of form factors, in contrast, the absolute phases of single form factors are not measurable.

A study based on dispersion relations gives the unique opportunity to explore the relationship between time-like and space-like behaviors, enabling the investigation of the form factors at negative q^2 , space-like region, and hence the acquisition of insights into dynamic and static properties of baryons.

As mentioned in Ref. [4], the form factor ratio G_E/G_M is the privileged observable to be studied via dispersive approaches, not only because it is completely known, but also because of its regular asymptotic behavior and its reality at well-known values of q^2 .

Thanks to the possibility of using such well-established theoretical constraints, relevant results can be obtained with a few experimental data, even a single point could play a crucial role.

On the other hand, the absence of data makes the predictions quite uncertain, particularly for the asymptotic behavior of the phase. Indeed, the values of the phase at time-like well beyond the present highest available data point can give precise predictions concerning the space-like behavior.

Gathering additional data at different energy points would be essential to bolster the predictive power of dispersive approaches to reveal additional remarkable attributes of baryons. More precise data and covering a larger energy region would be a pivotal step forward in understanding the dynamics underlying the interaction of hyperons.

The main points of this argumentation are summarized in the second paragraph of the revised manuscript.

REFERENCES

- [1] Mangoni, A. Hadronic decays of the J/ψ meson. arXiv:2002.09675.
- [2] Ferroli, R. B., Mangoni, A. & Pacetti, S. The cross section of $e^+e^- \rightarrow \Lambda\bar{\Sigma}^0 + c.c.$ as a litmus test of isospin violation in the decays of vector charmonia into $\Lambda\bar{\Sigma}^0 + c.c.$ *Eur. Phys. J. C* **80**, 903 (2020).
- [3] Zyla, P. A. et al. Review of particle physics. *Prog. Theor. Exp. Phys.* **2022**, 083C01 (2022).
- [4] Mangoni, A., Pacetti, S. & Tomasi-Gustafsson, E. First exploration of the physical Riemann surfaces of the ratio G_E^Λ/G_M^Λ . *Phys. Rev. D* **104**, 116016 (2021).

Reviewer#3

The authors have taken my comments from the first review carefully into account. A minor comment about the 6th blue bin of Fig. 3 as the authors discuss in their reply. I wonder if there is a breakdown of the inherent linear assumption in error propagation for this bin. As the individually binned measurements here only act as a cross check this is not an issue for the overall result but the authors might nevertheless with toy MC studies or similar want to study if this uncertainty can be trusted.

I summary, fully support a publication in the current form of the manuscript.

Re: We checked the error propagation again and found there is no problem. The most possible reason is due to the limited statistics for each bin and the the fitted $\Delta\Phi$ is quite small by coincidence, even the relative uncertainty is large, for the 6th bin, which the error propagated to C_{xy} is small. Fortunately, these bins are independent with each other and this figure just to display the polarization as a function of $\cos\theta$ and does not hurt the results.

We sincerely appreciate for your comments and suggestions, and we are also gratitude for your support of this work.

REVIEWERS' COMMENTS

Reviewer #1 (Remarks to the Author):

The introduction to the paper has significantly improved. It is much clearer and more understandable. Some parts of the introduction could use some English tuning or improvement. Nevertheless, my main concern has been addressed.

Reviewer #2 (Remarks to the Author):

The authors have addressed the points I raised. There is still some room left for improvement:

Concerning 1) Now it is better explained and it thus clearer described what the authors mean. Good.

Concerning 2) The estimate of isospin violation is a bit naive.

In fact, one can consider $(m_d - m_u)/(m_u + m_d) \sim 1/3$, which seems to indicate large isospin violation. The true measure would be $(m_d - m_u)/\Lambda_{\text{QCD}} \simeq 1/100$. For the case at hand, the estimate $(m_d - m_u)/\sqrt{s}$ is not quite correct, as this number is scale- and scheme-dependent. I would propose $(m_d - m_u)/m_c$ as a better quantity. Still gives the suppression.

Concerning 3) Now it is much more clear how to embed this single point of G_E/G_M in a larger framework. Dispersion relations for the Λ - Σ^0 transition have been studied by Granados et al. (Eur.Phys.J.A 53 (2017) 6, 117) and Lin et al. (Eur.Phys.J.A 59 (2023) 3, 54). These works need to be quoted.

I still believe that the point on the CP violation is most appropriate for Nature Communications, but the authors have made some efforts to improve their manuscript and have made it more accessible for a wider

audience. I thus recommend publication, provided the above made remarks are accounted for.

Reviewer#1

Reviewer 1 (Remarks to the Author):

The introduction to the paper has significantly improved. It is much clearer and more understandable. Some parts of the introduction could use some English tuning or improvement. Nevertheless, my main concern has been addressed.

Re: We really appreciate for your valusble comments/suggestions for the improvement of the manuscript!

Reviewer#2

The authors have addressed the points I raised. There is still some room left for improvement:

Concerning 1) Now it is better explained and it thus clearer described what the authors mean. Good.

Concerning 2) The estimate of isospin violation is a bit naive. In fact, one can consider $\frac{m_d - m_u}{m_u + m_d} \sim \frac{1}{3}$, which seems to indicate large isospin violation. The true measure would be $\frac{m_d - m_u}{\Lambda_{QCD}} \simeq \frac{1}{100}$. For the case at hand, the estimate $\frac{m_u - m_d}{\sqrt{s}}$ is not quite correct, as this number is scale- and scheme-dependent. I would propose $\frac{m_d - m_u}{m_c}$ as a better quantity. Still gives the suppression.

Re: In accordance with your suggestion, we estimate the suppression factor with mass of charm quark instead of the center-of-mass energy at J/ψ peak. And the other related numbers are updated accordingly.

Concerning 3) Now it is much more clear how to embed this single point of G_E/G_M in a larger framework. Dispersion relations for the Lambda-Sigma0 transition have been studied by Granados et al. (Eur.Phys.J.A 53 (2017) 6, 117) and Lin et al. (Eur.Phys.J.A 59 (2023) 3, 54). These works need to be quoted.

Re: Many thanks for providing these two nice references! They were included in the references of the updated manuscript.

I still believe that the point on the CP violation is most appropriate for Nature Communications, but the authors have made some efforts to improve their manuscript and have made it more accessible for a wider audience. I thus recommend publication, provided the above made remarks are accounted for.